# Podophyllotoxin: Recent Advances in the Development of Hybridization Strategies to Enhance Its Antitumoral Profile

**DOI:** 10.3390/pharmaceutics15122728

**Published:** 2023-12-04

**Authors:** Carolina Miranda-Vera, Ángela Patricia Hernández, Pilar García-García, David Díez, Pablo Anselmo García, María Ángeles Castro

**Affiliations:** 1Laboratorio de Química Farmacéutica, Departamento de Ciencias Farmacéuticas, CIETUS, IBSAL, Facultad de Farmacia, Campus Miguel de Unamuno, Universidad de Salamanca, 37007 Salamanca, Spain; cmivedoef@usal.es (C.M.-V.); angytahg@usal.es (Á.P.H.); pigaga@usal.es (P.G.-G.); pabloagg@usal.es (P.A.G.); 2Departamento de Química Orgánica, Facultad de Ciencias Químicas, Universidad de Salamanca, 37008 Salamanca, Spain; ddm@usal.es

**Keywords:** podophyllotoxin, natural products, chemomodulation, hybrids, cytotoxicity

## Abstract

Podophyllotoxin is a naturally occurring cyclolignan isolated from rhizomes of *Podophyllum* sp. In the clinic, it is used mainly as an antiviral; however, its antitumor activity is even more interesting. While podophyllotoxin possesses severe side effects that limit its development as an anticancer agent, nevertheless, it has become a good lead compound for the synthesis of derivatives with fewer side effects and better selectivity. Several examples, such as etoposide, highlight the potential of this natural product for chemomodulation in the search for new antitumor agents. This review focuses on the recent chemical modifications (2017–mid-2023) of the podophyllotoxin skeleton performed mainly at the C-ring (but also at the lactone D-ring and at the trimethoxyphenyl E-ring) together with their biological properties. Special emphasis is placed on hybrids or conjugates with other natural products (either primary or secondary metabolites) and other molecules (heterocycles, benzoheterocycles, synthetic drugs, and other moieties) that contribute to improved podophyllotoxin bioactivity. In fact, hybridization has been a good strategy to design podophyllotoxin derivatives with enhanced bioactivity. The way in which the two components are joined (directly or through spacers) was also considered for the organization of this review. This comprehensive perspective is presented with the aim of guiding the medicinal chemistry community in the design of new podophyllotoxin-based drugs with improved anticancer properties.

## 1. Introduction

Cancer has become a major public health problem [1]. According to the WHO [2], following ischemic heart disease, this malignancy is the second leading cause of death, accounting for nearly 10 million deaths in 2020. There are many available treatments for this disease, such as surgery, radiation therapy, chemotherapy, and so on. Chemotherapy is the most common method used for treatment despite severe side effects and a high percentage of treatment failures due to drug resistance. Development of new drugs could overcome such limitations [3,4]. In this sense, natural products play an important role in the discovery and development of chemotherapeutic agents [5].

Natural products have been invaluable as tools for deciphering the logic of biosynthesis and as platforms for developing front-line drugs [6]. They offer huge structural diversity and, in some cases, great biological potency. Additionally, their intricate molecular frameworks offer a range of uncharted molecular structures for the discovery of chemical probes and drugs [7,8]. In fact, among the new chemical entities approved by the FDA throughout the last four decades, 48.5% of them were natural products, semisynthetic natural products analogues, or synthetic products based on natural products’ pharmacophores [9]. In the anticancer area, plant-derived agents such as vincristine, paclitaxel (Taxol^®^), docetaxel, and topotecan are among the most effective cancer chemotherapeutics currently available [10].

Among the natural products, lignans stand out for their interesting properties [11]. Podophyllotoxin (**1**) (Figure 1) is one of the most representative compounds of this family. It is isolated from the rhizomes of species of the genus *Podophyllum* (Berberidaceae) [12], and it has been used since ancient times as a laxative and as a remedy for various medical complications, such as gonorrhea, tuberculosis, menstrual disorders, psoriasis, dropsy, cough, syphilis, and venereal warts [13]. Several different pharmaceutical properties have also been described for podophyllotoxin derivatives, such as myotoxic, neurotoxic, insecticidal, antimicrobial, anti-inflammatory, antispasmodic, hypolipidemic, immunosuppressive, analgesic, and cathartic activities [12]. Furthermore, podophyllotoxin has other uses in dermatology: it is a useful agent in psoriasis vulgaris and has also proved to be effective in the treatment of rheumatoid arthritis [14]. The antiviral activity of podophyllotoxin was also investigated, and it was found that podophyllotoxin was the most active component in inhibiting the replication of measles and the herpes simplex type I virus. In fact, this compound is indicated in several pharmacopeias as an antiviral agent in the treatment of *condyloma acuminatum* caused by the human papilloma virus [15].

The most notorious activity of podophyllotoxin is its antitumor activity [14,15]. However, its severe side effects, such as bone marrow suppression, hair loss, neurotoxicity, and gastrointestinal toxicity [11], have limited its development as anticancer agent. Nevertheless, podophyllotoxin has become a good lead compound for the development of new derivatives with fewer side effects and better selectivity. Nowadays, some semisynthetic derivatives, such as etoposide (**2**), teniposide (**3**), or etopophos (**4**) (Figure 1), are in clinical use to treat a variety of cancers, including small cell lung cancer, germ-line malignancies, leukemias, lymphomas, and sarcomas such as Kaposi’s sarcoma [14,16,17,18]. These semisynthetic derivatives differ from podophyllotoxin in the antitumor mechanism of action. While the antitumor activity of podophyllotoxin is based on the inhibition of tubulin polymerization into microtubules, causing cell cycle arrest in the metaphase [15], etoposide and analogs block the activity of the topoisomerase II enzyme (Topo II), stabilizing the Topo II-DNA cleavage complex and preventing relegation of the DNA double strand during replication [19,20].

Despite the importance of these podophyllotoxin derivatives in chemotherapy, their clinical use is also associated with some secondary effects related to drug resistance, cytotoxicity toward normal cells, and poor bioavailability [16,17]. Thus, the search for less toxic podophyllotoxin derivatives is an active area of research to overcome these side effects associated with current cancer treatments.

In recent years, novel agents have been synthesized in an attempt to overcome podophyllotoxin’s and etoposide’s limitations, such as GL331 (**5**), TOP53 (**6**), azatoxin (**7**), F14512 (**8**), NK611 (**9**), and tafluposide (**10**), among others (Figure 2). GL331 has proven to be effective against colorectal cancer; whereas TOP53 is effective against metastatic lung cancer and has shown higher antitumor activity than etoposide; F14512, on its part, exploits the polyamine transport system to target tumor cells. NK611 has also proven to have better antineoplastic activity than etoposide in various in vitro and in vivo investigations, and tafluposide exhibits a potent cytotoxic action thanks to its ability to inhibit both DNA topoisomerases I and II [16].

The important biological properties described for podophyllotoxin since its isolation in the mid-1950s justify that this compound and its derivatives have been and continue to be the objective of extensive chemical modifications in order to obtain better and safer anticancer drugs. For all this, recent reviews have focused on different aspects of history [12], synthesis or semisynthesis [21], and biological properties [11,21,22,23,24] of podophyllotoxin analogs.

In a similar fashion to other research groups, our group has been working for years on the chemomodulation of podophyllotoxin, and we have obtained good results regarding selectivity and immunosuppressive activity [25,26,27]. Now, as a new contribution to the field, this review focuses on the recent chemical modifications performed on the cyclolignan skeleton of podophyllotoxin, with special emphasis on hybrids or conjugates of podophyllotoxin with other natural products or different moieties that contribute to improve its bioactivity.

Hybridization, or molecular conjugation, is a drug discovery approach based on rational design that combines two or more molecules of natural or synthetic origin, giving rise to “hybrid systems” or “conjugates” [28]. These new drugs are synthesized in order to improve the pharmacological profile of the original molecules by modulating pharmacokinetics, transporting across membranes, or avoiding metabolic degradation. It can also be used to achieve synergistic effects between the two hybrid components or even to overcome acquired drug resistance to the components separately [29,30].

The concept of hybridization is therefore a broad and versatile concept for drug design. It is worth noting that there are different approaches from a structural point of view for the synthesis of the new molecular hybrids. The most commonly used strategies for the synthesis of new molecular hybrids are shown in Figure 3. The keys to these approaches involve the chemical bonding of the two components (molecular approach) or the role that each conjugate fragment plays in the final activity of the hybrid (pharmacological approach) [31].

Designing hybrid molecules based on natural products has become an effective strategy for obtaining new antitumoral agents with lower toxicity than traditional single-target-molecule co-administration methods, and it can also improve dose compliance and reduce drug interactions [32]. Thus, some hybrids of podophyllotoxin and other natural compounds have been synthesized to combine the great properties of the two chemical entities that form the hybrid derivate.

This review covers the literature published during the last six years (2017–mid-2023) considering the chemical modifications performed mainly at the C-ring (but also modifications at the lactone ring and at the trimethoxyphenyl pendant ring) together with the biological properties described for them. Through this review, we analyzed the published data with the purpose of guiding the medicinal chemistry community in the design of novel podophyllotoxin-based drugs with improved anticancer properties.

Most of the literature dealing with podophyllotoxin-related compounds used a numbering system derived from that for the naphthalene that forms the skeleton rings B and C. However, to be consistent with our previous works, in this review we follow the IUPAC recommendations for the lignan family [33], which was based on the two phenylpropanoid subunits that biogenetically formed the natural product. Thus, the phenylpropanoid unit included in the tetracyclic moiety is numbered from 1 to 9. The other phenylpropanoid unit is numbered from 1′ to 9′ as shown in Figure 4.

## 2. C-Ring Modifications of the Podophyllotoxin Skeleton

C-ring modifications of the podophyllotoxin skeleton are quite important. Structure–activity relationship (SAR) studies have demonstrated that some substitutions at C7 can be tolerable. However, aromatization of the C-ring, which leads to chirality loss, affords derivatives that are either less potent or completely inactive [14,16]. On another note, the antimitotic activity of podophyllotoxin is actually influenced by the C7 position. The free hydroxyl group at C7 contributes to the cytotoxicity, and its loss does not elicit significant variations in the antitumor activity since deoxypodophyllotoxin also binds to tubulin [14]. Epimerization at the C7 position gives rise to epipodophyllotoxin, which is less potent against the depolymerization of tubulin but increases the inhibitory activity against DNA-Topo II [14]. In addition, glycosylation at C7β can suppress tubulin polymerization inhibition activity. The introduction of bulky groups at C7β can enhance the antitumor activity of podophyllotoxin caused by inhibition of DNA-Topo II [34]. Thus, in recent years, several different substituents have been introduced at that position either in α- or β-disposition; these are presented below. They are grouped depending on the structural nature of the substituent introduced as natural-product derivatives, heterocyclic systems, and other compounds.

### 2.1. Hybrids of Podophyllotoxin with Different Natural Products

Among the natural products that have been recently hybridized with podophyllotoxin, secondary metabolites such as flavonoids, coumarins, pterostilbenes, or alkaloids have been considered along with primary metabolites such as retinoic acid, bile acids, sugars, or vitamins. Ester or amide bonds have been used to combine both fragments starting either from podophyllotoxin or its analogues epipodophyllotoxin and 7β-aminopodophyllotoxin.

#### 2.1.1. Podophyllotoxin Hybrids Synthesized through a Haloacyl Intermediate

Some hybrids of podophyllotoxin with formononetin, pterostilbenes, coumarins, and indarubicin are shown in Figure 5. They were synthesized from podophyllotoxin itself in two steps. First, a haloacyl intermediate was obtained via reaction with a chloro(bromo)acyl chloride followed by a substitution reaction of the halogen atom with a nucleophilic oxygen or nitrogen atom of the other natural compound under basic conditions, as stated in Figure 1.

Following the above procedure, a series of podophyllotoxin–formononetin derivatives (**11**) were synthesized by Yang et al. [35]. Formononetin is an *O*-methylated isoflavone widely present in legumes, many species of clovers, and the traditional Chinese herb *Astragalus membranaceus* and whose structure mimics the activity of endogenous estrogens. It has a wide range of pharmacological activities, such as antiproliferative activity, growth inhibitory activity, vasorelaxant action, a neuroprotective or cardioprotective effect, and mammary gland proliferative activity, as well as antiapoptotic, antioxidant, anti-inflammatory, and antimicrobial activities. In addition, the anticancer properties of formononetin have been well documented against several types of cancer, such as breast, colon, glioma, osteosarcoma, and lung cancer, among others [36,37,38]. The antiproliferative activity of the new podophyllotoxin–formononetin hybrids was evaluated against several tumor cell lines, showing IC_50_ values at micromolar and nanomolar levels. Compound **11a** (Figure 5) was the most promising compound, showing more sensitivity against the human lung carcinoma (A549) cell line than parental compound podophyllotoxin, with an IC_50_ value of 0.8 µM, which was 2.5-fold more potent than podophyllotoxin itself (IC_50_ = 1.9 µM). This derivative could also induce apoptosis in A549 cells through caspase-8 suppression and induce depolymerization of the microtubule network in these cells [35].

Pterostilbene is a natural demethylated analogue of resveratrol originally isolated from the heartwood of sandalwood with potential health benefits in inflammatory dermatoses, photoprotection, antioxidant activity, insulin sensitivity, blood glycemia and lipid levels, cardiovascular diseases, aging, memory, and cognition; it also exerts effects in cancer prevention and therapy in a wide range of tumors [39,40,41]. Pterostilbene was used by Zhang’s group to synthesize novel podophyllotoxin–pterostilbene hybrids (**12**) [42]. Conjugate **12a** (Figure 5) showed a better anticancer effect against uveal melanoma cells (MuM-2B cells) than pterostilbene (IC_50_ = 0.08 µM and 40 µM, respectively) but less cytotoxicity against human umbilical vein endothelial cells (HUVEC cells) than podophyllotoxin (IC_50_ = 0.36 µM and 0.01 µM, respectively), which indicates that this hybrid had less toxicity than the parent compound. Compound **12a** also had effects on the cell cycle since it arrested the MUM-2B cell cycle in the S cell cycle phase and promoted apoptosis in these cells. Furthermore, conjugate **12a** showed great potential in inhibiting migration and metastasis of MuM-2B cells by increasing E-cadherin levels and downregulating the levels of vascular endothelial growth factor receptor 2 (VEGFR-2) and matrix metalloproteinase-2 (MMP-2) in these cells. Reductions in the expression levels of Topo IIα and Topo IIβ enzymes were also observed [42].

Two podophyllotoxin–coumarin conjugates (**13**) were synthesized by Bai et al. [43]. Coumarins are heterocyclic moieties widely distributed in nature, and they have exhibited multiple pharmacological properties, such as antioxidant, antibacterial, antimicrobial, antiviral, hepatoprotective, and anti-inflammatory effects, apart from being chemopreventive and anticancer agents in vitro [44]. Of the two synthesized hybrids (Figure 5), compound 13b showed significant cytotoxic activity against human oral squamous carcinoma HSC-2, SCC-9, and A-253 cells with IC_50_ values of 0.22, 0.23, and 0.25 µM, respectively. Additionally, this compound showed less toxicity than podophyllotoxin against human renal tubular epithelial HK-2 cells (IC_50_ values of 0.62 and 0.04 µM, respectively). It was able to induce depolymerization of the microtubule network, which caused cell cycle arrest in the G2 phase, and it could also induce apoptosis through decreasing the mitochondrial membrane potential in HSC-2 cells [43].

New podophyllotoxin–indirubin conjugates (**14**) were synthesized by Wang and colleagues [45] using the procedure described in Figure 1. Indirubin, a bisindole alkaloid present in various plants, marine mollusks, and bacteria and detected in human urine, possesses several pharmacological activities like antipsoriatic, anti-Alzheimer’s, antiautoimmune, neuroprotective, anti-inflammatory, and anti-influenza activity. It also has antiproliferative activity, since it has been used for the treatment of chronic myelocytic leukemia by physicians in China for ages [46,47,48]. Among all the podophyllotoxin–indirubin derivatives synthesized (Figure 5), compound **14a**, with an acetyl group as linker, displayed the most potent antiproliferative effect against myeloblastic leukemia K562 cells and myeloblastic leukemia’s vincristine resistant cells (K562/VCR cells) with IC_50_ values of 0.034 and 0.076 µM, respectively. Other derivatives with extended-length linkers were found to have lower activity than **14a**, indicating that the linker length was critical to their antiproliferative effects. Compound **14a** could also induce cell apoptosis through the mitochondrial pathway, block the cell cycle in the G2 phase in K562/VCR cells, and inhibit tubulin polymerization in a similar way to podophyllotoxin [45].

#### 2.1.2. Podophyllotoxin Hybrids Synthesized via Direct Coupling with Natural-Product Derivatives Bearing a Carboxylic Group

An ester bond can be formed via the reaction of podophyllotoxin with a carboxylic acid in the presence of different coupling reagents, such as 1-(3-dimethylaminopropyl)-3-ethylcarbodiimide hydrochloride (EDCl), *N*,*N*′-dicyclohexylcarbodiimide (DCC), or *O*-(7-azabenzotriazol-1-yl)-*N*,*N*,*N*′,*N*′-tetramethyluronium hexafluorophosphate (HATU), as outlined in Figure 2. Additionally, similar coupling reagents have been used to join podophyllotoxin to other natural products through an amide bond using 7β-aminopodophyllotoxin as the starting compound. The carboxylic acid can be an inherent part of the other natural product, as in the retinoic acid hybrid (**15**), or be included as an additional substituent as in the case of carboline derivatives (**21**). These conjugates are depicted in Figure 6.

All-*trans* retinoic acid (ATRA), a metabolite of vitamin A, has great anticancer properties. It could inhibit cell proliferation and induce apoptosis markers in non-small-cell lung carcinoma in addition to being of use in the treatment of various tumoral diseases like Kaposi’s sarcoma or ovarian carcinoma [49,50]. Following the previous scheme and taking into account the great anticancer properties of all-*trans* retinoic acid, Zhang’s group synthesized a new podophyllotoxin–ATRA derivative (**15**, Figure 6) [51]. This compound showed significant antiproliferative activity against the human gastric cell lines MKN-45 and BGC-823 with IC_50_ values of 0.42 and 0.20 µM, respectively. It could also block the MKN-45 cell cycle at the G1 phase and the BGC-823 cell cycle at the G2 phase by decreasing the levels of CDK1, CDK2, cyclin A, and cyclin B1. Apoptosis was also altered in these cells when they were treated with this compound, which produces an increment in the number of apoptotic cells in both cell lines by enhancing the levels of cleaved caspase-3, -8, and -9 [51].

Some podophyllotoxin-linked β-carboline congeners (**21a**–**q**) that act as DNA-Topo II inhibitors were synthesized by Sathish and colleagues [52]. β-Carbolinic alkaloids are widely distributed in nature, being found in several families of plants, but they have also been found in cigarette smoke, overcooked foods, and wine. β-Carbolines have a wide spectrum of action, especially on the muscular, cardiovascular, and central nervous systems, and some reports indicate that β-carbolines have effective antioxidant properties [53]. Additionally, small molecules with a β-carboline scaffold possessed a tricyclic planar structure with suitable features for DNA interaction and enzyme inhibition activities like Topo II inhibition [54]. In order to obtain the podophyllotoxin-linked β-carboline derivatives, it was necessary to first synthesize β-carboline-3-carboxilic acids (**20**) from L-tryptophan (**16**) that was transformed into the corresponding methyl ester (**17**). Next, Pictet–Spengler condensation of compound **17** with different substituted benzaldehydes afforded methyl β-carboline-3-carboxilates (**18**, **19**). Finally, compound **19** was hydrolyzed to the free carboxylic acids (**20**), as described in Figure 3.

The final synthesis of the new β-carboline–podophyllotoxin derivatives was carried out following the procedure described above in Figure 2 starting from 7β-aminopodophyllotoxin. Among all the derivatives synthesized (**21a**–**q**), compounds **21i** and **21j** (Figure 6) showed promising cytotoxic activity with IC_50_ values of 1.1 µM against prostate cancer DU 145 cells. Cell cycle analysis and DNA-Topo II inhibition assays were carried out with these compounds. They showed that these conjugates blocked the DU 145 cell cycle at the S and G2/M phases and also could interfere with the catalytic activity of DNA-Topo II, acting as catalytic inhibitors [52].

Sometimes the carboxylic acid group is introduced through a spacer attached first to podophyllotoxin or to the other natural product, as shown for aurone and bile acid derivatives (Figure 7 and Figure 4).

Aurones are secondary metabolites of plants that belong to the family of flavonoids. They have been used since ancient ages as natural dyes. In recent years, aurones have gained attention because of their biological activities, including anti-inflammatory, antibacterial, antifungal, antimalarial, antileishmanial and anti-Alzheimer’s activities. However, the most important property of these metabolites is their anticancer activity, showing low toxicity and a broad spectrum of anticancer mechanisms [55,56]. In order to synthesize 4β-acetamidobenzofuranone–podophyllotoxin hybrids (**22**–**Ia**–**d** and **22**–**IIa**–**b**) (Figure 7), Paidakula and colleagues [57] first synthesized (*Z*)-2-benzylidenebenzofuran-3(2*H*)-one acid derivatives that were coupled with 7β-aminopodophyllotoxin, as indicated in Figure 2. Among all the derivatives synthesized, compound **22**–**Id** showed the most promising antiproliferative activity against the MCF7 and MDA-MB-231 human breast cell lines, A549 human lung cell line, and DU 145 human prostate cell line (IC_50_ values = 0.13, 0.45, 0.10 and 0.97 µM, respectively) [57].

Similar approaches were followed to obtain hybrids of podophyllotoxin with bile acids or ligustrazine derivatives through different amino acid linkers. The amino acids that acted as linkers were first attached to podophyllotoxin through an ester bond and then to the other natural fragment through an amide bond.

Bile acids are a group of steroid-based molecules biosynthesized from cholesterol in the liver and in the gall bladder. Not only are they involved in glucose, lipid, and energy homeostasis, but these compounds also function as signaling molecules to regulate metabolism, the inflammatory response, and gut microbiota. In addition, bile acids are known to regulate cell growth and proliferation. Dysregulation of bile acid homeostasis and signaling can have an impact on liver regeneration and tumorigenesis [58,59]. Furthermore, they can act as drug carriers and enhance the stability and biocompatibility of drugs in the body without causing side effects [60]. Owing to the great properties of bile acids and based on the results obtained in a previous report [61], De-Sheng’s group synthesized new podophyllotoxin-linked bile acid derivatives (**24** and **27**, Figure 4) attached directly or through ω-amino acid linkers, respectively [62]. For the latter, they first synthesized the acylated-*N*-Boc-protected derivatives of podophyllotoxin using *N*-Boc-aminocarboxylic acid carbon chains of different lengths and the corresponding coupling reagents. Deprotection of the amino group led to derivatives (**26**), and subsequent reaction with bile acids (**23**) led to the corresponding hybrids (**27**), as shown in Figure 4. They used six different bile acids: cholic, deoxycholic, chenodeoxycholic, ursodeoxycholic, hyodeoxycholic and lithocholic. The antitumor activity of the 30 new hybrids was evaluated against some human cancer cell lines. The hybrid with hyodeoxycholic acid and 6-aminocaproic acid as the linker (n = 5, R_1_ = R_3_ = H, R_2_ = α-OH) presented the most potent activity against the HepG2 cell line (IC_50_ = 0.18 µM) and also showed great selectivity against this tumor cell line. This compound could also induce early apoptosis and arrest the HepG2 cell cycle in the S phase [62].

In a similar fashion, Wu and colleagues synthesized several podophyllotoxin–α-amino acid–ligustrazine derivatives (**30**, Figure 5) [63]. Ligustrazine, or tetramethylpyrazine, is an alkaloid monomer isolated and extracted from the Chinese herbal medicine *Lisgusticum wallichii* that has various pharmacological activities, such as anticardiovascular, antiplatelet, ischemic stroke, anti-Alzheimer’s, neuroprotective, and anticancer effects. It has also been reported to suppress caspase-3 activation, increase Bcl-2 expression, and diminish Bax expression in some brain tissues [64,65,66]. To join ligustrazine to podophyllotoxin, one of the methyl groups was first oxidized to the corresponding acid and then attached to the podophyllotoxin skeleton using different L-α-amino acids as linkers. For that, the *N*-Boc-protected amino acids were first attached to podophyllotoxin through an ester bond to obtain **28**. The ligustrazine moiety was then joined to the deprotected amino acids derivatives (**29**) via an amide function, affording hybrids (**30**) (Figure 5). The antiproliferative activity of the new derivatives was tested against three tumor cell lines. Among all the tested compounds, the deprotected podophyllotoxin–amino acid derivatives (**29**) were more potent than the corresponding protected ones (**28**), and the podophyllotoxin–amino acid–ligustrazine derivatives (**30**) showed better selectivity against tumor cell lines. Interesting results were obtained with the intermediate **28b**, which has a *N*-Boc-*N*-methyl glycine. Such derivative showed a similar antiproliferative activity to the *N*-Boc-deprotected compound (**29b**) toward the A549 tumor cell line, but it also showed less cytotoxicity toward L-02 normal cell lines. Compound **28b** could also induce apoptosis and cell cycle arrest in the S phase in the A549 cell line [63].

#### 2.1.3. Podophyllotoxin Hybrids with Other Natural Products Attached through a Triazole Ring

Although later in this review, several podophyllotoxin derivatives bearing a triazole ring in C7 position are considered separately, we have included some hybrids in this section that used such a heterocycle as the linker that joined podophyllotoxin and other natural compounds. They were obtained via click chemistry-type reactions, specifically the Huisgen 1,3-dipolar cycloaddition between a propargyl intermediate of a natural compound and an azido intermediate of podophyllotoxin (Figure 6). Thus, the heterocycle was built at the same time as the hybrid was formed. Briefly, the propargyl rest was joined to the non-lignan natural compound via a reaction with propargyl bromide, and the azido intermediate (**31**) was synthesized using sodium azide (NaN_3_) in the presence of trifluoroacetic acid (TFA). Then, the coupling reaction between the two intermediates was carried out following the Huisgen 1,3-dipolar cycloaddition conditions, as shown in Figure 6. Further information on the triazole scaffold and Huisgen 1,3-dipolar cycloaddition conditions can be found below in the section on podophyllotoxin–heterocycle hybrids (Section 2.2.1).

Several podophyllotoxin hybrids, such as podophyllotoxin–epigallocatechin-3-gallate (EGCG) conjugates (**32**), podophyllotoxin–camptothecin conjugates (**33**), a podophyllotoxin–curcumin hybrid (**34**), or bispodophyllotoxins (**35**–**37**), have been synthesized through this strategy; these are shown in Figure 8.

Podophyllotoxin–EGCG derivatives (**32-I** and **32-II**, Figure 8) were synthesized by Zi and colleagues using the procedure shown in Figure 6 [67,68]. EGCG is the dominant active polyphenol of green tea (*Camellia sinensis*). It possesses antioxidant, antitumor, and anti-inflammatory features [69]. Several studies have demonstrated that the anticancer actions of this catechin include antiproliferative, pro-apoptotic, antiangiogenic, and anti-invasive functions and multiple antitumor mechanisms, such as alteration in the tumor cell cycle and production of epigenetic changes in gene expression [69,70]. The novel podophyllotoxin–EGCG derivatives exhibited good antiproliferative activity against all cancer cell lines tested. Among all the methylated EGCG–podophyllotoxin hybrids, compound **32-Ia** (R_1_ = R_2_ = CH_3_) showed the most potent cytotoxicity against A549 cells with an IC_50_ value of 10 µM. Among the non-methylated EGCG–podophyllotoxin conjugates, compound **32-IIb** (R_1_ = R_2_ = H) showed the most potent antiproliferative activity against the A549 cell line, even better than the methylated conjugate **32-IIa**, with an IC_50_ value of 2.2 µM. Docking studies of these EGCG methylated and non-methylated conjugates suggested that such compounds could have a dual mechanism of action: inhibition of tubulin polymerization via binding at the colchicine binding site and inhibition of Topo II [67,68].

Camptothecin is a plant alkaloid present in wood, bark, and fruits of the Asian tree *Camptotheca acuminata*. It was discovered in 1966 by Wall and Wani in a systematic screening of natural products for anticancer drugs, and then in 1985 it was identified as a specific inhibitor of the enzyme DNA-topoisomerase I (Topo I) [71]. This drug reversibly induces single-strand breaks, thereby affecting a cell’s capacity to replicate. Camptothecin also stabilizes the so-called cleavable complex between Topo I and DNA [72]. As is known, some derivatives of podophyllotoxin, like etoposide, are DNA-Topo II inhibitors. So, the main purpose of the synthesis of camptothecin–podophyllotoxin conjugates was the development of compounds that can act on both Topo I and Topo II enzymes. The conjugates synthesized by Li et al. [73] (**33a**–**b**, Figure 8) were tested against a panel of five cancer cell lines: leukemia (HL-60), hepatoma (SMMC-7721), lung cancer (A549), breast cancer (MCF7), and colon cancer (SW-480). It was found that both derivatives showed weak activity against the cancer cell lines tested, and only compound **33b** displayed significant anticancer activity against the HL-60 cell line with an IC_50_ value of 18 µM [73], suggesting that 4′-*O*-demethylation is necessary for the activity of these hybrid compounds.

On their part, Duan and colleagues synthesized two novel tetrahydrocurcumin-podophyllotoxin hybrids (**34a**–**b**, Figure 8) following the same protocol described in Figure 6 [74]. Tetrahydrocurcumin is one of the major metabolites of curcumine, the main compound of the rhizomes of *Curcuma longa* (turmeric). However, it also exists naturally in the roots of *Curcuma zeoaria*, *Zinger mioga*, and *Zinger officinale*. Regarding its pharmacological properties, it has been recognized as anticancer, antiaging, and an antioxidant, and it can also prevent neurodegenerative disorders [75,76,77]. The new tetrahydrocurcumin–podophyllotoxin derivatives were tested against some human cancer cell lines, with both (**34a**–**b**) showing promising anticancer activity against the HCT 116 and HeLa cell lines. Particularly, compound **34a**, with IC_50_ values of 18 and 83 µM, respectively, was four times more potent against HCT 116 than against HeLa cells [74].

Dimeric podophyllotoxin derivatives synthesized by Zi et al. (**35a**–**b**, **36a**–**f**, and **37a**–**h**) [78] are shown in Figure 9. They include two triazole moieties joined through a previously dipropargyl functionalized glycerol or glucose scaffold that allowed the synthesis of the triazole moieties simultaneously. The derivatives were evaluated for their cytotoxicity against the same five human cell lines (HL-60, SMMC-7721, A549, MCF7, and SW-480). The IC_50_ values obtained revealed that most of these derivatives were nearly inactive (IC_50_ > 40 µM). However, compound **36c** was active against all five cancer cell lines with IC_50_ values ranging from 0.43 to 3.5 µM. SAR studies suggested that the spacer between the two podophyllotoxin moieties can largely affect their activity [78].

Biotin, or vitamin H, is a water-soluble vitamin that belongs to the vitamin B complex and is an essential nutrient in the living organism, although it cannot be synthesized by higher organisms, in which a dietary intake is needed. Biotin is found in a large number of foods, including eggs, bread, green-leaf vegetables, or beef liver. In eukaryotic cells, biotin functions as a prosthetic group of enzymes, collectively known as biotin-dependent carboxylases, which catalyze key reactions in gluconeogenesis, fatty acid synthesis, and amino acid catabolism. Furthermore, recent evidence suggests that biotin also plays unique roles in cell signaling, the epigenetic regulation of genes, and chromatin structure [79,80]. Nonetheless, despite that biotin and other vitamins are necessary for the survival and division of normal cells, biotin requirement in tumor cells is higher than in normal cells due to their rapid growth and division. In addition, recent studies have shown that some cell lines express higher levels of biotin receptors than normal cells, such as L1210RF (leukemia), Ov2008 (ovarian), Colo–26 (colon), P815 (mastocytoma), M109 (lung), RENCA (renal), and 4T1 (breast) [81,82]. Due to the importance of biotin in cancer, several biotinylated podophyllotoxin derivatives (**42**–**45**, Figure 7) were synthesized by Zi and colleagues [83]. Three different series of derivatives were synthesized: ester-linked derived from epipodophyllotoxin (**42a**–**b**) and from podophyllotoxin (**43**), amide-linked obtained from 4β-aminopodophyllotoxin (**44a**–**d**), and triazole-linked derivatives (**45a**–**d**). The latter were synthesized first by converting podophyllotoxin into the 7β-azide derivative **31** followed by reaction with propargyl alcohol to obtain the triazole derivative **40**. Three series of compounds were synthesized via the reaction of 6-biotinylaminocaproic acid (**41**) with the corresponding podophyllotoxin derivates **38**–**40** (Figure 7).

The cytotoxicity of these conjugates was evaluated in several cancer cell lines. The results suggested that compounds **42** and **43**, with an ester linkage, exhibited better antiproliferative activities with IC_50_ values ranging from micromolar to nanomolar. They also suggested that the 6-aminocaproic acid linking spacer had a variable influence on the cytotoxicity. Among these derivatives, compound **42a**, with the biotin directly attached to podophyllotoxin, showed the most promising antiproliferative activity, even higher than podophyllotoxin in hepatoma and colon cancer cell lines (the SW480 and SMMC-7721 cell lines; IC_50_ 0.23 and 0.56 µM for **42a**; 4.1 and 9.4 µM for podophyllotoxin). Also, compound **42a** could induce cell apoptosis in the lung cancer cell lines NCI-H1299 and H1975 through an increment in the expression levels of cleaved-caspase-3 and cleaved-PARP in a dose-dependent manner. It was also able to significantly suppress the growth of S180 tumor xenografts in mice [83].

Finally, we have included in this section several podophyllotoxin–polyamine hybrids despite their synthesis prior to the time period covered by this review [84,85] due to the fact that a study about their activity against different Topo II isoforms was published in 2018 in order to compare their activity with that of etoposide. The importance of etoposide in cancer treatment is well known. However, the drug is associated with very severe side effects, one of the most dramatic of which is the generation of treatment-related acute myelogenous leukemia [85]. To minimize off-target effects, polyamine–podophyllotoxin conjugates were evaluated by Oviatt and colleagues [86]. Among them, compound **8**, called F14512 (Figure 2 and Figure 10), with a tetramine spermine side chain that was a candidate for phase I clinical trials [87], had the highest ability to induce double-stranded breaks in DNA and turned out to be a much stronger Topo II poison than etoposide. Moreover, in vitro studies of this compound against different Topo II isoforms revealed that it is more potent against the Topo IIβ isoform than to the α-isoform [86]. Additionally, this compound showed strong therapeutic efficacy against spontaneous non-Hodgkin lymphoma in a randomized double-blind trial in dogs that justified its evaluation in human clinical trials [88].

#### 2.1.4. Podophyllotoxin Glycoconjugates

Among the primary metabolites, carbohydrates have been of interest in cancer therapy for over a century [89] since Otto Warburg observed differences in the 1920s between cancer cells and normal cells in glucose metabolism. Cancer cells tend to depend on glycolysis for acquiring energy even in aerobic conditions, and this fact was later named the “Warburg effect”. Therefore, cancer cells usually have a higher expression of glucose transporters, which the scientific community has used to facilitate the entry of carbohydrate-conjugated antitumor drugs into cells [90].

Glycoconjugates that act as anticancer drugs contain sugar moieties that allow the specific binding of the antitumor drugs with those specific glucose transporters. Glucose transporters transfer the drugs to the tumor cells, where they are selectively uptaken and thus the organ toxicity thus is minimized. This has become an attractive strategy in drug design in order to improve drug efficacy and pharmacokinetics and to reduce side effects [91,92].

The podophyllotoxin glycoconjugates are not new, since etoposide and teniposide, two of the semisynthetic derivatives of podophyllotoxin that are in clinical use, include a sugar moiety in their structure. The introduction of a β-glycosidic moiety at C7 position of podophyllotoxin together with the 4′-*O*-demethylation is related to the change in the mechanism of action of these compounds, as mentioned in the Introduction. Instead of being inhibitors of tubulin polymerization like their parent compound podophyllotoxin, they are potent DNA-Topo II inhibitors [7,15].

Teniposide is less frequently used for chemotherapy in comparison with etoposide, although it has shown higher in vitro DNA-Topo II inhibition than etoposide [16]. New derivatives of teniposide have been synthesized in recent years by Cheng and colleagues [93] by introducing different fluorobenzoheterocyclic derivatives into glucose positions 2 and 3 in teniposide using different carbodiimides as coupling reagents (Figure 8). Among them, only compounds **48i** and **48j** presented a similar antitumor activity to teniposide against the HepG2, HeLa, A549, and MCF7 tumor cell lines, but they presented a reduction in the toxicity by around 10 times against HL-7702, H8, MRC-5, and HMEC human cells compared to teniposide. Cell cycle analysis and apoptosis studies revealed that these derivatives could block the cell cycle in the G2/M phase like teniposide, and they also could induce apoptosis in HepG2 cells, but in a weaker manner than etoposide. DNA-Topo II inhibitory activity was also measured, and the results indicated that compounds **48i** and **48j** inhibited this enzyme in a concentration-dependent manner [93].

In the last six years, other podophyllotoxin glycoconjugates have been synthesized by Zi and colleagues, such as the per-butyrylated and per-acetylated glycoside derivatives **49** and **51** (Figure 9) [94,95]. The final glucoside conjugates were synthesized by coupling the tetra-acetylated residue of glucose (**54**) or the tri-butyrylated residue of xylose (**56**) with podophyllotoxin/demethylpodophyllotoxin. The corresponding deacylated analogues **50** and **52** with free glucose or xylose residues were also obtained. At this point, we would like to notice that during deprotection, isomerization of the lactone ring must have taken place under the strong basic conditions used. The authors claimed that the *trans*-lactone ring of the podophyllotoxin skeleton is maintained in compounds **50** and **52**. However the chemical shift reported for C9′ in the ^13^C NMR spectra did not indicate that, since it appeared at over 175 ppm in all the compounds, which is characteristic of *cis*-lactones [96].

The per-acetylated glycoside was synthesized first by treating *D*-glucose (**53**) with acetic anhydride and sodium acetate. Then, 1,2,3,4,6-penta-*O*-acetyl-α/β-glucopyranoside was treated with ammonia to get the free hydroxy group at the anomeric position, obtaining 2,3,4,6-tetra-*O*-acetyl-α/β-*D*-glucopyranoside (**54**) in this manner. The tri-butyrylated residue of xylose (**56**) was synthesized in a similar manner through the treatment of *D*-xylose (**55**) with butyric anhydride and iodine followed by ammonia treatment (Figure 10).

Novel glucoside and xyloside derivatives, per-acylated or not, were evaluated for their antiproliferative activity against the HL-60, SMMC-7721, A549, MCF7, and SW480 cancer cell lines. In both cases, the per-acylated derivatives were more cytotoxic than the derivates with the free sugar rests, as expected for *cis*-lactones; among them, the 4′-*O*-demethylated derivatives (compounds **49b** and **51b**) showed the highest cytotoxicity against the five cancer cell lines tested. Thus, the per-butyrylated xyloside (**49b**) showed the highest cytotoxicity for the HL-60 cell line with an IC_50_ value of 2.9 µM and for the A549 cell line with an IC_50_ value of 4.4 µM [95]. In the case of the corresponding per-acetylated glucoside (**51b**), it showed IC_50_ values ranging from 3.2 to 11 µM; the highest cytotoxicity and selectivity were against the SW480 colon cancer cell line (IC_50_ = 3.3 µM; Selectivity Index = 6.7) [94].

During the time covered by this review, other podophyllotoxin glycoconjugates that used 1,2,3-triazole as a linker between podophyllotoxin and the sugar residues have been synthesized following the conditions outlined at Figure 6. The use of this strategy is exemplified by the 1,2,3-triazole-linked picropodophyllin glycoconjugates synthesized by Zi et al. [97] containing maltose, 1,6-β-*D*-diglucose, or several other disaccharides and trisaccharides, in which the sugar residues were also joined by triazole rings (**57I**-**V**, Figure 11). Most of the compounds synthesized showed weak cytotoxicity with IC_50_ values > 40 µM against the cell lines tested, as expected for picropodophyllin derivatives. However, compound **57II** with a 1,6-β-*D*-diglucose residue showed strong anticancer activity against all cancer cell lines tested (HL-60, SMMC-7721, A549, MCF7, and SW480) with IC_50_ values ranging from 0.67 to 7.4 µM [97].

The podophyllotoxin glycoconjugates **59a**–**l** obtained by Nerella and colleagues [98] were synthesized through a click chemistry reaction between 4β-*O*-propargylated podophyllotoxin (**59**) and sugar azides, although in this case, silver (I)- *N*-heterocyclic carbenes were used as catalysts instead of copper salts (Figure 12). Compound **59a**, with a tetraacetylglucosyl residue, was the most promising of the series against the DU 145 prostate cancer cell line with IC_50_ values of 1.0 µM. Compound **59e**, bearing a tetraacetylgalactosyl residue, was the most promising against the MCF7 cell line with an IC_50_ value of 1.3 µM. Docking studies of compounds **59a** and **59e** revealed that both derivatives showed a high binding affinity toward DNA-Topo II and that they were in good agreement with observed IC_50_ values [98].

### 2.2. Podophyllotoxin Derivatives with Heterocyclic Systems

Heterocyclic compounds, aromatic or not, are a significant source of pharmacologically active scaffolds, which can be prepared synthetically or isolated from natural sources [99]. An important feature of heterocycle bioactive compounds is the presence of constituent heteroatoms (mainly nitrogen, sulfur, and oxygen in many combinations) that directly affect the reactivity of the skeleton, the bioactivity of the compounds, and the interactions between biological targets and target modulators [100]. Therefore, their significance in drugs rests on the diversity and versatility of substitution patterns that these rings can undergo.

In this section, heterocyclic substituents at podophyllotoxin’s C7 position are considered and grouped according to the heterocyclic size: five- or six-membered rings and benzoheterocycles with one, two, or three heteroatoms. In some cases, the heterocycle is joined directly to the podophyllotoxin without any linker. In other cases, the linker is only one heteroatom, and in most of the cases, they are connected by linkers with different length chains through ester or amide bonds. Also, the heterocycle could be introduced as such or formed at the last reaction step.

#### 2.2.1. Five-Membered Heterocycles with Two Heteroatoms

The imidazole nucleus is commonly found in numerous endogenous biomolecules, including the neurotransmitter histamine and the amino acid histidine, among others. Apart from that, the imidazole unit is present in a great variety of medicinal agents and shows several pharmacological activities, such as antibacterial, antitumor, antitubercular, analgesic, anti-VIH, and antifungal [101,102,103].

Cao and his colleagues synthesized the *N*-acyl imidazole derivative of podophyllotoxin GMZ-1 (**60**, Figure 11) [104]. Compound **60** showed high antiproliferative activity against the K562 and K562/A02 cell lines with IC_50_ values of 0.08 and 0.12 µM, respectively. In addition, this compound was able to induce apoptosis in the K562/A02 cell line in a time- and concentration-dependent manner, which indicated its potentiality to overcome leukemia cell resistance to apoptosis induced by chemotherapy [104].

On their part, Yin and colleagues have synthesized new podophyllotoxin derivatives bearing nitrogen-containing heterocycles, such as imidazole, methylimidazole, or 1,2,4-triazole [105]. With the aim of synthesizing these derivatives, they first created a reaction of podophyllotoxin with 2-chloropropionyl chloride (**61**) following the procedure described previously in Figure 1. Then, compound **62** reacted with imidazole, 1,2,4-triazole, and 2-methylimidazole to obtain compounds **63**–**65**, whose treatment with various bromides generated the subsequent podophyllotoxin imidazolium/ triazolium salts (compounds **66**–**68**), as shown in Figure 13.

The cytotoxic activity of the new conjugates was evaluated against some human cancer cell lines, including HepG2, A549, MDA-MB-231, and HCT 116 cells. All the imidazole and 1,2,4-triazole derivatives **66**–**68** showed an excellent cytotoxic activity against all cancer cell lines tested. Among them, compound **66f**, bearing a 2-naphthylmethyl substituent at the third position of the imidazole ring, showed the most potent antitumor activity with IC_50_ values of 0.07, 0.29, 0.11, and 0.04 against the HepG2, A549, MDA-MB-231, and HCT 116 tumor cell lines, respectively. On another note, compound **66f** could also induce apoptosis in the HCT 116 cell line and arrest the cell cycle of this tumor cell line in the G2/M phase [105].

The thiazole nucleus is a very important heterocycle in many biologically active compounds, and it is also a fascinating building block in medicinal chemistry, since a wide range of activities have been described for compounds containing thiazole rings, like antioxidant, anti-inflammatory, antimicrobial, antifungal, antiviral, anticonvulsant, neuroprotective, and antitumor activities [106,107].

Sang and colleagues synthesized 7β-thiazole derivatives of 4′-*O*-demethylepipodophyllotoxin (**71a**–**i**) linked only by a nitrogen atom, as shown in Figure 14 [108]. Firstly, the corresponding 2-(2-aminothiazol-4-yl)acetic esters or amides (**70**) were synthesized from ethyl 2-(2-aminothiazol-4-yl)acetate as previously described [109]. 4′-*O*-Demethyl-7β-iodoepipodophyllotoxin (**69**) on its part was synthesized by using methanesulfonic acid and sodium iodide and then coupled to the heterocycle in the presence of triethylamine. The cytotoxic activities of the nine obtained derivatives (**71a**–**i**) were evaluated against four human cancer cell lines (A549, HepG2, HeLa, and LOVO).

The results showed that the new compounds had greater or equal antitumor activity against these cancer cell lines, and 2-aminothiazol-4-yl esters **71g** and **71h** appeared to be more potent than the other derivatives synthesized. Their effect on DNA-Topo II and double-strand breaks was also studied, showing that compound **71h** (R = OCH_2_CH_3_ and n = 0) decreased the expression of both Topo IIα and Topo IIβ in the A-569 cell line in a dose-dependent manner. Furthermore, it induced supercoiled pBR322 DNA relaxation to form linear DNA [108].

Another heterocycle with two heteroatoms that has been attached to podophyllotoxin is isoxazoline. The isoxazoline scaffold is found in several compounds exerting a wide range of biological activities and pharmacological properties, occupying a prominent position in psychotropic drugs, but it also serves as a valuable precursor for the construction of diverse molecules, including natural products [110,111].

Yang and colleagues synthesized novel isoxazoline-containing podophyllotoxin/2′,6′-dihalogenopodophyllotoxin derivatives (**78 I**–**IVa**–**e**, Figure 15) [112] with improved insecticidal/acaricidal activities compared to other previously synthesized dihalogenopodophyllotoxin derivates [113,114]. They were synthesized using 1,3-dipolar cycloadditions between dihalogenopodophyllotoxin acrylates (**77**) and nitride oxide dipoles (**74**) obtained through the corresponding oximes (**73**), as shown in Figure 15.

Among all the derivatives, compounds **78 IIc** and **78 IIIc**, bearing 4-chlorophenyl as isoxazoline substituent and chlorine atoms in the C6′ or C6′ and C2′ positions of podophyllotoxin, respectively, exhibited good insecticidal and acaricidal activities that were even better than the toosendanin used as a reference but not a higher cytotoxicity. This fact demonstrates that the introduction of the halogen atom/atoms at the C2′, C6′ position of podophyllotoxin and the introduction of chlorine atom at the C4 position on the phenyl substituent of the isoxazoline fragment were very important for their insecticidal and acaricidal activities [112].

Zefirov and colleagues also synthesized several podophyllotoxin and epipodophyllotoxin derivatives bearing substituted isoxazole fragments during the time covered by this review [115]. They first synthesized the isoxazole unit from ethyl-2,4-dioxopentanoate (**79**) following a known procedure [116,117,118], as shown in Figure 16. Then, bromination, iodination, and nitration at position 4 of the isoxazole ring was carried out following several procedures previously described [119,120,121]. The final derivatives (**81**–**I**–**a**–**d** and **81**–**IIa**–**d**) were obtained by coupling isoxazolecarboxylic acids **80a**–**d** with podophyllotoxin and epipodophyllotoxin using coupling reagents.

A primary screening of the cytotoxicity of these compounds was conducted using a test based on determining the survival of co-cultivated modified cell lines of tumor and non-tumor etiology (A549, EGFP, and VA13_katushka). The results showed that cell treatment with such compounds at a concentration of 100 nmol/L for 72 h caused a decrease in the amount of both the tumor and non-tumor cell lines by 40–60% with IC_50_ values in the range of nanomolar. However, the authors could not establish whether the introduction of an halogen atom in the fourth position of the isoxazole ring could be a key feature in the cytotoxicity of the compounds [115].

#### 2.2.2. Five-Membered Heterocycles with Three Heteroatoms

In this section, several triazole, oxadiazole, and thiadiazole heterocycles are considered.

Under the generic name of triazole, two heterocycles are possible: 1,2,3-triazole and 1,2,4-triazole. The most frequent is 1,2,3-triazole, which is a fundamental building block in different bioactive compounds because it is remarkably stable toward hydrolysis, oxidative/reductive conditions, and enzymatic degradation [122,123].

1,4-Substituted-1,2,3-triazoles have multiple roles in bioactive molecules: as a basic pharmacophore element, they participate in the formation of hydrogen bonding or hydrophobic interactions; as a molecular scaffold, they regulate other pharmacophore elements to maintain an active conformation; as a linker group, they link conjugates, molecules, or probes [124]. Triazoles can also show bioisosteric effects on peptide linkage, the aromatic ring, double bonds, and the ubiquitous bioheterocycle imidazole ring. Consequently, these central structural motifs of biological interest have manifested therapeutical potential in different domains: antitumor, anti-inflammatory, analgesic, anti-HIV, anticandidal, antiparasitic, etc.; they have also been used as various enzyme inhibitors, such as histone deacetylase, alkaline phosphatase, etc. [122].

Due to the importance of the triazole skeleton in medicinal chemistry, in the period covered by this review, several podophyllotoxin hybrids in which the triazole ring acts as a linker have been synthesized through the click chemistry strategy. A few examples of podophyllotoxin conjugates with other natural products obtained with this strategy were described in the previous Section 2.1.3. The concept of click chemistry, first introduced by Sharpless, was used to describe a synthetic strategy to join organic and bioorganic molecules. Click chemistry refers to the set of highly efficient, reliable, and stereoselective reactions that can be used to develop promising structures using facile reaction conditions and easily available starting materials [125]. Among all the type of click reactions, Huisgen 1,3-dipolar cycloaddition of alkynes and azides yielding triazoles is the most representative [126]. In the case of the conjugates previously described in this review, the triazole ring was placed directly at podophyllotoxin’s C7 position, acting as the linker between the two natural-product moieties. In this section, the compounds shown below present the triazole ring separated from the podophyllotoxin skeleton by ester or amide bonds. They were synthesized by coupling an azido derivative of podophyllotoxin with an alkyne derivative of an aliphatic or aromatic compound using Cu (I) in its salt form (Cu_2_SO_4_·5H_2_O) as a catalyst, as indicated in Figure 17.

The 7α-triazolacetyl–podophyllotoxin derivatives (**82a**–**w**, Figure 12) synthesized by Hou and colleagues [127] from podophyllotoxin following the method described in Figure 17 exhibited low nanomolar IC_50_ values against the three cancer cell lines tested (A549, MCF7, and prostate cancer PC-3). Compound **82u** (Figure 12) was the most representative of the phenyl-triazolo derivatives of the series, presenting the greatest antiproliferative activity against all the cell lines (IC_50_ values = 30.1 µM for A549 cells, 21.1 µM for PC-3 cells, and 25.2 µM for MCF7 cells). It could also induce apoptosis in PC-3 cells in a dose-dependent manner through an increment in the level of cellular ROS and blocking the cell cycle in PC-3 cells at the G2/M phase [127].

Vishnuvardhan et al. synthesized novel 7β-[(4-substituted)-1,2,3-triazol-1-yl]acylaminopodophyllotoxin derivatives (**83a**–**l**, Figure 12) from 7β-aminopodophyllotoxin [128]. Compounds **83c** and **83f**, bearing substituted phenyl derivatives, were able to increase their cytotoxicity against HeLa cells compared to podophyllotoxin with IC_50_ values of 0.9 and 0.07 µM, respectively. These compounds could also induce apoptosis in these cells and arrest HeLa cells cell cycle in the G2/M phase. Furthermore, compound **83f** could increase its cytotoxic activity against MCF7 and HT-29 compared to podophyllotoxin with IC_50_ values of 0.9 and 0.1 µM, respectively [128].

Several 7β-triazolamides of podophyllotoxin (**85a**–**t**, Figure 18) with the ability to cause DNA-Topo IIα inhibition were synthesized by Reddy and colleagues [129]. In this case, the triazole ring was built before coupling it with podophyllotoxin. First, a substituted triazolic acid (**84**) was synthesized that subsequently was coupled with 7β-aminopodophyllotoxin (**38**), as shown in Figure 18. Among the 20 derivatives synthesized, compounds **85b**, with a 4-methoxyphenyl substituent, and **85g** and **85i**, with halogen-phenyl substituents, exhibited significant cytotoxicity with IC_50_ values of less than 1 µM against all the cell lines tested (HeLa, MCF7, DU 145, HepG2, and HT-29). Moreover, compound **85g**, with a 4-chlorophenyl substituent, was the most promising compound in the series with IC_50_ values in the range of 0.70–4.11 µM. A DNA-Topo II inhibition assay confirmed that compounds **85b**, **85g**, and **85i** inhibited the activity of Topo II [129].

Apart from the imidazole derivatives of podophyllotoxin described previously (Figure 13), Yin and colleagues also synthesized 1,2,3-triazole derivatives of podophyllotoxin using a click chemistry protocol (Figure 19). For this, they created a reaction of podophyllotoxin with thionyl chloride, obtaining 7-chlorinated podophyllotoxin (**86**). Then, a nucleophilic substitution was carried out with sodium azide to obtain compound **87**, and finally, the Huisgen 1,3-dipolar cycloaddition was performed using various terminal alkynes. All the 1,2,3-triazole derivatives (**88a**–**e**) except compound **88e**, bearing a naphthalene ring at the fourth position of the triazole ring, showed IC_50_ values in the range of 0.04–9.64 µM against the human tumor cell lines tested (the HepG2, A549, MDA-MB-231, and HCT 116 cell lines) [106].

As far as we know, only one example with 1,2,4-triazole heterocycle joined directly to podophyllotoxin was found during the period covered by this review. In this case, the triazole was attached to podophyllotoxin through a sulfur atom. Zhao’s group previously synthesized the antitumor agent 7β-(1,2,4-triazol-3-ylthio)-7-deoxypodophyllotoxin (**90**) by reacting podophyllotoxin with 1,2,4-triazole-3-thiol (**89**), as shown in Figure 20 [130]. In 2017, they published the results obtained in the cytotoxic studies of this derivative against the HeLa, BGC-823, and A549 tumor cell lines [131]. Compound **90** showed IC_50_ values (IC_50_ = 0.17, 0.34, and 0.79 µM, respectively) lower than those of colchicine and podophyllotoxin (IC_50_ = 33.10, 21.75, and 35.08 µM for colchicine and 8.12, 15.41, and 13.64 µM for podophyllotoxin for the HeLa, BGC-823, and A549 tumor cell lines, respectively). Also, they were able to crystalize its tubulin complex, and the interactions at the α,β-tubulin interface were studied [131].

On another note, among the five-membered heterocycles with three different heteroatoms, several examples of podophyllotoxin derivatives bearing a 1,3,4-oxadiazole ring were found. This five-membered heterocycle has gained a lot of attention in pharmaceutical sciences in recent years since it is a commonly used pharmacophore for drug design due to its metabolic profile and ability to engage hydrogen bonding with receptor sites [132]. Also, this heterocycle ring is a very good bioisostere of amides and esters. 1,3,4-Oxadiazole-based compounds have shown interesting pharmacological properties, including antitumor, tyrosinase inhibitor, antibacterial, antifungal, antitubercular, antimalarial, muscle relaxant, anticonvulsant, antihypoglycemic, antioxidant, insecticidal, and anti-VIH activity [133,134].

The synthetic strategies to introduce this heterocycle on the podophyllotoxin skeleton are similar to those described before for the synthesis of other heterocycles. Researchers either built the ring at the last step of the synthetic procedure or coupled the cyclolignan with the previously formed heterocycle. Thus, Ren and colleagues, during the period covered by this review, prepared several novel podophyllotoxin derivatives with a 1,3,4-oxadiazole heterocycle (Figure 21) by first converting podophyllotoxin into 7β-aminopodophyllotoxin (**38**). The amino group was then transformed into the corresponding isothiocyanate (**92**) through a dithiocarbamate intermediate (**91**). Finally, reaction with several hydrazides followed by treatment with TsCl yielded the final oxadiazole derivatives (**93a**–**i**), as shown in Figure 21 [135].

The most interesting derivative was **93b**, with a methyl substituent, which caused a strong inhibition of the survival of HepG2 tumor cell lines in a dose- and time-dependent manner when they were treated with this compound. In addition, compound **93b** could arrest the cell cycle of HepG2 cells in the S phase in a time-dependent manner through the suppression of the expression of cyclin D1 and a decrease in the expression of CDK2 and CDK4 in a dose-dependent manner. Regarding apoptosis, this podophyllotoxin derivative could produce significant apoptosis in HepG2 cells through the activation of the mitochondria-based intrinsic apoptosis pathway [136].

Wang and colleagues and Han et al. both obtained 1,3,4-oxadiazole and 1,3,4-thiadiazole derivatives of podophyllotoxin using an alternative path of synthesizing the heterocycle before it joining to podophyllotoxin [137,138]. First, they synthesized 1,3,4-oxa(thia)diazole-acid intermediates (**97**) from benzoate derivatives, as shown in Figure 22.

The final derivatives **98** and **99** were synthesized via a reaction with podophyllotoxin using coupling reagents (Figure 22).

Among all the 1,3,4-oxadiazole derivatives synthesized by Wang and colleagues (Figure 13) [137], compound **98c**, with a 4-methoxyphenyl substituent, exhibited the most potent antiproliferative activity against the four cancer cell lines, but it specifically had better antiproliferative activity against the MCF7 cell line with an IC_50_ value of 2.54 µM. Further studies were carried out with this compound, and the results showed that it could induce cell cycle arrest in the G2/M phase in a dose-dependent manner in MCF7 cells via an increment in the expression of the proteins cyclin A2 and CDK2. Compound **98c** could lead to tubulin depolymerization in MCF7 cells [137].

At the same time, Han and colleagues synthesized both 1,3,4-oxadiazole and 1,3,4-thiadiazole derivatives of podophyllotoxin and compounds **99m** and **99i**, belonging to the 1,3,4-oxadiazole series and bearing an halogen atom; these were the most notorious derivatives (Figure 13) [138]. Compound **99m** showed better anticancer activity than podophyllotoxin against MDA-MB-231 cells (IC_50_ = 7.28 µM), while compound **99i** displayed better antiproliferative against MCF7 cells (IC_50_ = 2.46 µM). The antiproliferative activity of Compound **99i** was further investigated in other breast cancer cell lines such as the BT474, SK-BR-3, and MDA-MB-453 cell lines, exhibiting excellent antiproliferative activity in all the breast cancer cell lines tested (IC_50_ values = 5.23, 4.48, and 8.23 µM, respectively). In addition, compound **99i** could arrest the MCF7 cell cycle in the G2/M phase by increasing the protein levels of cyclin B1 and CDK1 and also could induce apoptosis by increasing the levels of proapoptotic proteins [138].

#### 2.2.3. Six-Membered Heterocycles

During the period included in this review, piperazine and pyridine were the six-membered heterocycles used by researchers to synthesize new podophyllotoxin derivatives.

Pyridine-based ring systems are one of the most abundant heterocycles in nature and one of the most widely used in the field of drug design. In biological systems, pyridine-based systems take part in redox reactions where NAD reduces its pyridine into dihydropyridine, rendering NADH; in plants, they are mostly found in alkaloids [139]. In addition, pyridine derivatives are known for their antimicrobial, antiviral, antioxidant, anti-inflammatory, antidiabetic, antimalarial, and antipsychotic activities [140].

Wang’s group, on its part, has synthesized new derivatives of podophyllotoxin with 2-aminopyridines [141]. They were synthesized by reacting podophyllotoxin or 4′-*O*-demethylepipodophyllotoxin with HBr, obtaining in this manner a 7β-bromide intermediate that experimented with nucleophilic substitution via 2-aminopyridines to yield the desired podophyllotoxin derivatives (Figure 23).

The cytotoxic activity of these pyridine derivatives was evaluated against several tumoral cell lines. Among all the derivatives, those containing a fluorine atom (compounds **101c** and **101d**) were, in general, the most cytotoxic of the series with an IC_50_ under the micromolar level (IC_50_ values = 0.20, 2.18, 4.49, 9.04, and 26.23 µM for **101c** and 0.06, 0.96, 1.42, 6.65, and 19.63 µM for **101d** against the HeLa, BGC-823, MCF7, A549, and Huh cell lines, respectively). Furthermore, cell cycle arrest studies and apoptotic studies were carried out to elucidate the mechanisms of the new derivatives. The cell cycle studies showed that the treatment of HeLa cells with **101c** and **101d** increased the percentage of cells at the G2/M phase, but interestingly, the change in the fluorine with a methyl group in **101a** did not induce cell cycle arrest significantly at that phase of the cell cycle. These results showed that fluorine derivatives exhibited the most potent cytotoxic activity against HeLa cells compared to the other derivatives. Regarding the apoptotic studies, **101c** and **101d** induced apoptosis in a concentration-dependent manner through an increase in p53 protein levels [141].

Considering the great importance of derivative **101c** (R_1_ = CH_3_; R_2_ = F), further studies were carried out with this compound in order to investigate its biological effects on the microtubule and in cell cycle- and apoptosis-related proteins [142]. These studies revealed that when cells were treated with different concentrations of **101c**, they displayed a disorder in the microtubule distribution and that the microtubule network completely disappeared, with all cells becoming roundish, showing evident apoptosis. Furthermore, it inhibited the phosphorylation of MDM2, an ubiquitin ligase of p53 [142].

Compounds **107** and **108** are novel dithiocarbamates derivatives obtained by Li et al. bearing one and two pyridine rings, respectively (Figure 24B) [143]. Both were prepared by reacting 4′-*O*-demethylepipodophyllotoxin with the corresponding dithiocarbamates (**104** and **106**) that were synthesized from (pyridin-2-yl)methanal or di(pyridin-2-yl)methanone, as shown in Figure 24A [144].

Compound **107** displayed significant inhibitory activity against HepG2 cells and HCCLM3 cells with an IC_50_ of 3.89 and 11.01 µM, and it also inhibited migration and invasion ability of HCCLM3 by reducing the expression levels of metalloproteinase 2 and 9 (MMP-2 and MMP-9). Furthermore, treatment of HepG2 cells with this compound induced the apoptotic ratio of these cells caused by the upregulation of the expression levels of cleaved-caspase 3 and Bax protein and the downregulation of the expression levels of Bcl-2 protein [145].

Compound **108**, on its part, exhibited significant antiproliferative activity against three hepatocellular carcinoma cell lines (the HepG2, Bel-7402, and HCCLM3 cell lines) with IC_50_ values < 3 µM. Additionally, **108** inhibited cell migration and invasion in the HCCLM3 cell line by reducing both MMP-2 and MMP-9 in a similar manner to **107**. Furthermore, **108** could trigger the upregulation of the p53 protein and induce cellular ROS production [143].

In recent years, several piperazine–podophyllotoxin derivatives have been synthesized due to the notorious importance that the piperazine scaffold has gained lately in drug design. Piperazine is one of the most versatile six-member-containing heterocycles. The *N*-4 nitrogen of piperazine can be used as a basic amine, while hydrogen bond acceptors and hydrophobic groups can be easily introduced at the *N*-1 of nitrogen without the addition of a stereocenter. Piperazine is present in several natural bioactive or synthetic derivatives, and it is one of the major components of the plethora of clinically useful compounds like antiemetics, antihelminthics, antiallergic anxiolytics, antipsychotics, antidepressants, and so on [146,147,148,149]. As with other heterocycles, the piperazine ring can be attached directly to the C7 position or through a spacer.

Piperazine-acetyl–podophyllotoxin ester derivatives (**110a**–**p**, Figure 25) were synthesized by Xue et al. by joining the acid intermediates of piperazine **109** with podophyllotoxin in the presence of coupling agents [150]. Most of the obtained derivatives showed antitumor activity against the cell lines tested. Compound **110e**, bearing a 3,4-dichlorophenyl substituent, displayed the most potent inhibitory activity against the MCF7 cell line with an IC_50_ value of 2.78 µM. Cell cycle studies of this compound showed that treatment of MCF7 cells with **110e** led to G2/M phase arrest in a dose-dependent manner due to the phenotypic changes in the skeleton network of tubulin that triggered tubulin depolymerization and an increment in the expression levels of the protein CDK1 [150].

Zhang and colleagues and Wu et al. reported new piperazine–podophyllotoxin derivatives in which the piperazine scaffold was introduced directly at podophyllotoxin’s C7β position [151,152]. First, they synthesized a 7α/β-iodo derivative of podophyllotoxin and demethylepipodophyllotoxin that was subsequently treated with the piperazine intermediates to afford 7α or β-piperazine podophyllotoxin derivatives, as shown in Figure 26. Additionally, the substituent at *N*-4 of the piperazinyl ring can also be introduced before or after the heterocycle is coupled with podophyllotoxin.

After obtaining 7-piperazinyl-podophyllotoxin (**111**), Zhang and colleagues synthesized the final products (**113**–**I** and **113**–**II**) via condensation of **111** with cinnamic acid derivatives (**112**) (Figure 27) [151]. The antiproliferative activity of the 12 synthesized derivatives was screened against some human cancer cell lines, with most of the compounds showing a remarkable increment in the cytotoxicity against the tumor cell lines. However, compounds with a 7α configuration (**113**–**I**) exhibited better cytotoxicity against the tumor cell lines than compounds with a 7β configuration (**113**–**II**). Compound **113**–**Ie**, with an α configuration and no substituent in the phenyl ring of cinnamic acid, showed the most promising antiproliferative activity against the HeLa cell line with an IC_50_ value of 0.14 µM. In addition, this compound was able to inhibit colony formation in all the cancer cell lines tested, especially in MCF7 cells, in which compound **113**–**Ie** could also reduce the mitochondria membrane potential, induce apoptosis by reducing the Bcl-2/Bax ratio, and induce depolymerization of the microtubule cytoskeleton [151].

#### 2.2.4. Benzoheterocycle Derivatives

Indole is one of the most biologically active, abundantly distributed heterocycles in nature; it is found in natural products such as alkaloids, plant and animal hormones, etc. Biological evaluations and mechanism of action studies of anticancer indoles revealed that these compounds target diverse pathways in cancer cells. In particular, a great number of small molecules containing an indole scaffold have been described as tubulin polymerization inhibitors with the potential of interacting with colchicine’s binding site. Other important targets for anticancer indoles are histone deacetylases, sirtuins, PIM kinases, or DNA topoisomerases [153].

Some podophyllotoxin–heterocyclic/benzoheterocyclic conjugates have been synthesized by coupling the C7 hydroxyl group from podophyllotoxin with the corresponding heterocyclic acid using coupling reagents, as shown in Figure 28.

Han and colleagues synthesized several new podophyllotoxin–benzoheterocyclic derivatives (**114a**–**j**, Figure 14) from podophyllotoxin and the corresponding benzoheterocyclic acid following the procedure described above (Figure 28) [154]. Among all these derivatives, compound **114d**, with an indole moiety, showed the greatest cytotoxicity with IC_50_ values of 1.93, 2.20, 5.94, and 10.10 µM against the HepG2, HeLa, A549, and MCF7 tumor cell lines, respectively. In addition, compound **114d** blocked the HepG2 cell cycle in G2/M caused by the upregulation of CDK1 protein expression and the downregulation of the expression levels of cyclins A, B, and D1. Additionally, compound **114d** induced apoptosis in HepG2 cells through the downregulation of the expression of Bcl-2 and the upregulation of the protein levels of Bad and Bax. Alteration in the tubulin cytoskeleton was also reported when HepG2 cells were treated with **100d** at 4 µM [154].

In a similar fashion, Zhang et al. synthesized some new indole esters of podophyllotoxin (**115a**–**j**, Figure 14) that might be accommodated deeply into the colchicine binding site of β-tubulin and simultaneously occupy the hydrophobic cavity of α-tubulin [155]. The 10 derivatives obtained were tested for their cytotoxicity against drug-sensitive and drug-resistant human leukemia cells (the K562 and K562/VCR cell lines). Compounds **115b**, **115h**, and **115i** showed better antiproliferative activities against K562 cells and K562/VCR cells with IC_50_ values of 0.04, 0.08, and 0.1 µM and 0.20, 0.23, and 0.22 µM, respectively. Cell cycle analysis of **115i** revealed that this compound could cause arrest of the cell cycle of K562/VCR cells in the G2 phase, and apoptotic analysis suggested that **115i** exerted its antiproliferative activity through the induction of cell apoptosis [155].

Finally, Zhao and colleagues, on their part, synthesized 7β-aminopodophyllotoxin derivatives (**116**, Figure 29) in which the benzoheterocycle was attached to the cyclolignan skeleton through a nitrogen atom by reacting 7β-iodopodophyllotoxin (**69**) with the corresponding amino-substituted indole, indazole, and quinolone derivatives (Figure 29) [156]. The 12 novel derivatives synthesized were assessed against four human tumor cell lines: HepG2, HeLa, A549, and MCF7; the results showed that most of the new congeners exhibited higher cytotoxic activities against the tumor cell lines than the parental compounds. In addition, compounds **116c** and **116f**, with 1*H*-indol-6-yl and 1*H*-indazol-5-yl substituents, reached nanomolar concentration levels (IC_50_ values of 0.1, 0.08, 0.08, and 0.07 µM for compound **116c** and 0.3, 0.2, 0.2, and 0.2 µM for compound **116f**). These compounds could also arrest the cell cycle of MCF7, HeLa, and A549 in the G2/M phase and increase the rate of apoptosis in a dose- and time-dependent manner in the MCF7 cell line [156].

Zhang et al. synthesized 18 new benzo-heterocyclic derivatives of podophyllotoxin from 4′-*O*-demethylepipodophyllotoxin and SH-containing benzo-five-membered heterocycles (**118a**–**r**, Figure 30) [157]. The antiproliferative activity of the new derivatives obtained was evaluated against the HepG2, HeLa, A549, and MCF7 cancer cell lines. All the compounds showed promising activity against the tumor cell lines tested. Among them, fluorobenzoxazole and fluorobenzothiazol derivatives **118b** and **118n** showed the best antiproliferative activity (IC_50_ values of 1.5, 1.5, 2.3, and 1.4 µM for compound **118b** and 1.4, 2.3, 3.4, and 1.2 µM for compound **118n** against HepG2, HeLa, A549, and MCF7 tumor cells, respectively) with low toxicity (millimolar level) on normal cells. Further studies of cell cycle arrest and apoptosis showed that they could block the cell cycle in HepG2 cells at the G2/M phase and improve their ability to induce apoptosis because they could inhibit microtubule polymerization and induce DNA double-strand breaks [157].

### 2.3. Hybrids of Podophyllotoxin with Other Moieties

Podophyllotoxin was also coupled to commercial drugs bearing a carboxylic acid moiety, such as non-steroidal anti-inflammatory drugs (NSAIDs) and the antitumoral drug lonidamine, by Zang’s group [158,159]. The hybrids were synthesized following the same procedure of coupling podophyllotoxin with the corresponding acids (aspirin, indomethacin, and ibuprofen as the NSAIDs and lonidamine), as shown in Figure 31.

Podophyllotoxin–NSAID conjugates (**119**–**121**, Figure 15) were synthesized with the purpose of overcoming cell multidrug resistance (MDR) [159]. The conjugates were tested against drug-sensitive and drug-resistant hepatocellular carcinoma cells (Bel-7402 and Bel-7402/5-FU), with all of them showing potent inhibitory effects on the proliferation of both types of cells. Notably, a podophyllotoxin–aspirin conjugate (compound **119**) showed the best antiproliferative activity against Bel-7402 and Bel-7402/5-FU (IC_50_ = 0.19 and 0.06 µM, respectively), and it also possessed selective cytotoxicity against the drug-resistant Bel-7402/5-FU cells with a resistance factor of 0.32, which suggested that the podophyllotoxin–aspirin hybrid had better potential than other podophyllotoxin derivatives to overcome drug resistance in Bel-7402/5-FU cells. Additionally, several studies of P-gp and MRP protein expression levels carried out via Western blotting indicated that the podophyllotoxin–aspirin conjugate significantly reduced the expression level of multidrug-resistance-associated protein 1 (MRP1) and slightly decreased the level of P-gp [159].

Lonidamine (LND, 1-(2,4-dichlorobenzyl)-1*H*-indazole-3-carboxylic acid), an antispermatogenic and antineoplastic agent, is an inhibitor of the glycolytic enzyme hexokinase II, which is upregulated in several tumors. It has low anticancer activity when used alone but presents selectivity to various tumors, so it is commonly used as a sensitizer to chemotherapeutic agents and physical therapy. Additionally, new targets for this compound have been found recently, such as some components of the mitochondrial electron chain and some mitochondrial transporters [160,161,162].

The antiproliferative activity of the podophyllotoxin–lonidamine hybrid (compound **122**, Figure 15) [158] was evaluated against the K562, K562/VCR, and K562/ADR cell lines, displaying stronger cytotoxicity against such cell lines when compared to etoposide, with IC_50_ values of 0.423, 0.578, and 0.560 µM, respectively. In addition, cell cycle and apoptosis analyses showed that compound **122** could arrest the K562/VCR cell cycle in the G2 phase and that the compound could increase the rate of K562/VCR apoptotic cells caused by the disruption in the mitochondrial function [158].

With the aim of overcoming multidrug resistance, new nitric oxide-donating podophyllotoxin derivatives (**123a**–**b**) were obtained by Zhang and colleagues (Figure 32) [163]. They were synthesized by coupling podophyllotoxin with chloroacyl chlorides, as stated for other hybrids, and then the acyl-intermediate was directly treated with silver nitrate (AgNO_3_) to produce the final compounds (**123a**–**b**). Both compounds **123a** and **123b** showed potent antiproliferative activity against three leukemic cell lines (K562, K562/VCR, and K562/ADR) with nanomolar IC_50_ values in the range of 0.007–0.08 µM in all cell lines tested, and they could also arrest the K562/ADR cell cycle in the G2 phase. Due to the presence of the -ONO_2_ moiety in the derivatives, the intracellular NO levels were investigated, and the results suggested that compound **123a** could induce an increment in the intracellular NO level in K562/ADR cells. Additionally, the impact of compound **123a** on the expression levels of MDR-related proteins in K562/ADR cells was determined via Western blotting, and it was found that derivative **123a** could significantly reduce P-gp levels. However, this compound displayed no significant effect on MRP1 expression [163].

A nitrosyl derivative was also created by Yang and colleagues [164]. 4β-[(2″,2″,6″,6″-Tetramethyl-1″-oxyl-piperidin-4″-yl)amino]-4′-*O*-demethylepipodophyllotoxin (**124**, Figure 16) is a new podophyllotoxin derivative spin-labeled at C7 position and whose synthesis was described in a previous report [165], but in the time covered by this review, new antiproliferative studies have been performed. Those studies revealed that **124** inhibited proliferation of different human gastric cancer cells at different differentiation degrees. For example, within gastric cancer cells at different degrees of proliferation, compound **124** was more potent against MKN28 cells from well-differentiated adenocarcinoma than HGC-27 cells from undifferentiated adenocarcinoma. Additionally, the compound decreased the expression of HSP90 in the BGC-823 and HGC-27 cell lines in a concentration-dependent manner and could induce apoptosis in these cells after 48 h of treatment [164].

Several prodrugs of podophyllotoxin that can be activated by NQO1 oxidoreductase were synthesized by Qu and colleagues (Figure 33 and Figure 34) [166]. They were constituted using podophyllotoxin as the active drug, a self-immolative linker, and a benzoquinone as a NQO1-responsive trigger group. NQO1 is an oxidoreductase enzyme that catalyzes the direct two-electron reduction for various quinones, and it is upregulated in some human cancer tissues, such as non-small-cell lung carcinoma or pancreatic carcinoma [167,168]. Compound **128** was synthesized following Figure 33. First, trimethylhydroquinone (**125**) reacted with 3,3-dimethylacrylic acid (3,3-DMAC), and the product (**126**) was then subjected to *N*-bromosuccinimide treatment to give quinone acid (**127**), which was then coupled to podophyllotoxin.

Other similar prodrugs with longer self-immolative linkers, including an aromatic ring, were also synthesized. As shown in Figure 34, connection to podophyllotoxin was achieved by means of a carbonate group (**130**). The synthesized prodrugs were tested against NQO1-overexpressing A549 and HepG2, hypoxia-induced A549 and HepG2, and Taxol-resistant A549 human cancer cell lines. Compound **131**–**II** (R = CH_3_) displayed the most potent activity and selectivity against NQO1-overexpressing A549 and HepG2 cells (IC_50_ = 0.1 and 0.08 µM, respectively) and had a significant inhibitory effect on A549/T cells, HepG2/Hyp, and A549/Hyp cell lines with low toxicity to normal cells. The in vitro podophyllotoxin release profile of compound **131**–**II** was investigated through an HPLC assay, showing that the release of the prodrug followed a time-dependent pattern [166].

Sun and colleagues synthesized 18 podophyllotoxin-β-aroylacrylic-related acids (**135a**–**m**, Figure 35) [169]. β-Aroylacrylic acids can be considered a structural combination of chalcones and cinnamic acids, two naturally occurring aromatic compounds that exert a broad spectrum of pharmacological activities that are mostly dependent on the position of the substituent groups [170,171]. β-Aroylacrylic acid derivatives (**134**) were synthesized from several acid anhydrides and substituted benzenes. Once the acid derivatives were obtained, they were coupled to podophyllotoxin using coupling reagents (Figure 35). The new derivatives exhibited significant toxicity against the three cancer cell lines tested and were more potent than etoposide. Compound **135l**, the one bearing 3-bromo aroylacrylic acid, showed the most potent antiproliferative activity in the series against the HGC-27 human gastric cell line with an IC_50_ value of 0.89 µM, and it also showed lower cytotoxic activity against the normal cell lines. In addition, this compound could arrest the cell cycle of HGC-27 cells in the G2/M phase caused by the upregulation of CDK1 protein expression and the downregulation of cyclin B1 and CDC25C. Apoptosis was also induced by compound **135l** in HGC-27 cells in a time-dependent manner through the mitochondrial pathway [169].

In recent years, it was discovered that ferrocenyl derivatives and their analogues are able to show novel and unexpected properties. For example, it has been demonstrated that a ferrocenyl analogue of paclitaxel is more potent in terms of its antiproliferative activity than the parental compound [172].

Hence, some podophyllotoxin–ferrocenyl conjugates were synthesized by Wieczorek and colleagues (Figure 36) [173]. Three types of conjugates were synthesized using podophyllotoxin and ϖ-ferrocenylalkanoic acids (**136**) and ϖ-alkynoylferrocenes (**137**) as starting materials to create compounds linked by ester or amide bonds and by a 1,2,3-triazole ring, respectively. Ester–ferrocenyl derivatives (**138**) were synthesized via the *O*-acylation of podophyllotoxin with ϖ-ferrocenylalkanoic acids (**136**) following a previously described procedure [174]. On their part, the aminoferrocenyl derivatives (**140**) were synthesized by coupling 7β-aminopodophyllotoxin (**38**) and the same ϖ-ferrocene carboxylic acids (**136**). To synthesize the triazolo–ferrocenyl conjugates (**139**), ϖ-alkynoylferrocenes (**137**) were first prepared via acylation of ferrocene with ϖ-alkynoic acids, and then they were coupled with 7β-azidopodophyllotoxin (**31**). The antiproliferative activity of these derivatives was tested against some human cancer cell lines. The results showed that the ester conjugates with a ferrocenyl moiety (**138**) were more potent than the 1,2,3-triazolo conjugates (**139**). Amide podophyllotoxin–ferrocenyl conjugates (**140**) were the least cytotoxic. The most active compounds of the series were **138c** and **138d** with IC_50_ values between 0.11 and 0.68 µM against the colorectal adenocarcinoma Colo-205, HCT 116, and SW620 cell lines and the A549, HepG2, and MCF7 cell lines [173].

With the purpose of improving podophyllotoxin’s interactions with DNA Topo IIα and overcoming its problems related to toxicity, Wei and colleagues synthesized several 7β-acetylamino-substituted podophyllotoxin derivatives (**142a**–**t**, Figure 37) [175] via a reaction of podophyllotoxin or 4′-*O*-demethylepipodophyllotoxin with chloroacetonitrile (ClCH_2_CN) instead of the recurrent 2-chloroacetyl chloride (Figure 37). Thus, 7β-chloroamido podophyllotoxin/demethylpodophyllotoxin intermediates (**141**) were formed directly. After that, such intermediates reacted with substituted amines to afford the final new derivatives (**142a**–**t**). The antiproliferative activity of the derivatives was tested against some human cancer cell lines. Compound **142g**, with a disubstituted *n*-propyl amine group, showed improved and selective toxicity against the HeLa and human bladder carcinoma (T24) cell lines with IC_50_ values of 2.34 and 2.98 µM, respectively. This compound could also induce apoptosis and cell cycle arrest in the G2/M phase in HeLa cells [175].

Several new ester derivatives of podophyllotoxin were obtained (**143a**–**t**, Figure 38) by Zhao et al. [176] through the combination of podophyllotoxin and *N*-Boc-amino acids and other moieties. The antiproliferative activity of the novel derivatives was evaluated against five human cancer cell lines (PC-3M, HEMEC, A549, MCF7, and HepG2 cells). Compound **143e**, with a Boc-glycine rest, exhibited the strongest antiproliferative activity against the PC-3 cell line with IC_50_ value of 1.28 µM. This compound could also induce apoptosis in this tumor cell line in a concentration-dependent manner [176].

Considering the importance of the sulfamate moiety in drug design, Bader and colleagues synthesized new podophyllotoxin–sulfamate derivatives (**144a**–**f**, Figure 39) [177]. The novel derivatives were synthesized from podophyllotoxin, reacting first with chlorosulfonic acid and then with ammonia or aryl/heteroarylamines. Compound **144b** of the series, bearing a 2-pyridylmethyl substituent, showed great antiproliferative properties against tumor cell lines such as the MCF7, A-2780, and HT-29 cell lines with IC_50_ values in the range of 0.15–0.22 µM, and it could also increase the apoptotic rate in MCF7 cells in a dose-dependent manner [177].

Yang et al., on their part, synthesized novel *N*-sulfonyl amidine derivatives of podophyllotoxin through a cooper-catalyzed sulfonyl azide–alkyne cycloaddition/ring cleavage (CuAAC/ring-opening reaction) [178]. This reaction was carried out in a one-pot fashion with 7β-aminopodophyllotoxin (**38**) or 4′-*O*-demethyl-7β-aminopodophyllotoxin, terminal alkynes, sulfonyl azides, CuI, and Et_3_N to afford the new derivatives (**145 Ia**–**v**; **145 IIa**–**f**, Figure 40). The antiproliferative activity of the new analogs was tested against the A549 cell line, with many of them showing moderate to good anticancer activity. Compounds **145**–**Ir**, **145**–**Iv**, and **145**–**IIc**, which possessed -CF_3_, 10-camphor, and -CF_3_ groups, respectively, showed the most promising antiproliferative activity, even more potent than the parent compound etoposide (IC_50_ values: 2.44, 5.21, 1.65, and 12 µM, respectively) [178].

Another interesting functional group used to join podophyllotoxin to other moieties is carbamate, which also has great properties in medicinal chemistry. The emerging role of carbamates in drug design is related to their chemical and metabolic stability, their capability to increase permeability across cellular membranes, and their ability to improve the biological activity of active pharmacophores of structurally different natural or synthetic compounds [179,180].

Hence, Xu and colleagues synthesized a new series of aminosulfonyl carbamates (**147a**–**l**, Figure 41) [181]. The derivatives were prepared by reacting podophyllotoxin with chlorosulfonyl isocyanate (CSI) to give the intermediate **146**, which subsequently reacted with aliphatic or heterocyclic amines to obtain *N*-(aminosulfonyl)-4-podophyllotoxin carbamates (**147a**–**i**). The cytotoxic activity of the new derivatives was evaluated, and compound **147e**, with a morpholine residue, showed the most promising antiproliferative activities with IC_50_ values of 0.5, 1.5, 16.5, and 0.7 µM for HeLa, A549, HCT-8, and HepG2 cells, respectively. Furthermore, cell cycle studies suggested that compound **147e** could inhibit the progression of the cell cycle in HeLa cells in the G2/M phase, and nuclear condensation studies showed that compound **147e** produced nuclear fragmentation, which meant that such compound could also induce apoptosis in HeLa cells [181].

Novel aliphatic, five-membered heterocyclic and benzoheterocyclic amino derivatives of podophyllotoxin were synthesized by Tian’s group [178]. Some of these derivatives could also be included in previous sections of this review; however, we decided to collect them together at this point because they were prepared through an imine intermediate formed between 7β-aminopodophyllotoxin (**38**) and several aldehydes [182]. The imine intermediate of podophyllotoxin (**148**) was reduced to the corresponding amine to yield the final compounds (**149a**–**k**, Figure 42). Some of the new derivatives showed better antiproliferative activities against the HeLa cell line than etoposide. However, the nature of the R_1_ substituent influenced the cytotoxic activity. Compounds with an aliphatic amine showed better cytotoxicity than derivatives with heterocyclic moieties. Also, benzoheterocyclic compounds showed lower activity than those with five-membered heterocycles [182].

Finally, several phenyl derivatives of podophyllotoxin were synthesized during the time covered in this review through multiple synthetic strategies.

Cui Hu et al. synthesized new podophyllotoxin–phenoxyacetyl derivatives (**151a**–**l**, Figure 43) by binding the corresponding phenoxyacetic acid derivatives (**150a**–**l**) [183]. Among the novel derivatives, compounds **151d** and **151i** exhibited the best antiproliferative effect against the HepG2 cell line (IC_50_ values of 5.21 and 4.92 µM, respectively), and compound **151d**, with a 2,5-dimehtylphenyl group, exhibited the best antiproliferative activity against the HeLa cell lines (1.64 µM), even better than podophyllotoxin. Furthermore, compound **151d** could block the HeLa cell cycle at the G2/M phase, which was caused by the suppression of Cdc25 protein levels [183].

On their part, Xi et al. synthesized new podophyllotoxin derivatives targeting DNA-Topo II (**154a**–**j** and **155a**–**o**), as shown in Figure 44 [184]. Briefly, after podophyllotoxin 4′-*O*-demethylation, displacement of the C7 hydroxyl group gave compound **153**, which was subsequently reduced via a Pd-C catalytic hydrogenation. Derivatives **154a**–**j** were then obtained through a condensation reaction between **153** and the corresponding acids. In the case of derivatives **155a**–**o**, the synthetic route was similar, but intermediate 7-azidoepipodophyllotoxin (**31**) was used instead. Among all the synthesized derivatives, compound **155i** with a 2-(*N*-benzyl-*N*-methylamino)ethyl group was the most effective against the human cancer cell lines tested with IC_50_ values between 4.1 and 10 µM and showing high potency against the HepG2 cell line with an IC_50_ value of 2.18 µM [184].

Related to the above derivatives, compound **156** was synthesized by Wang and colleagues (Figure 17); it had dimethylaminoacetamido as the phenyl substituent and was methylated at C4′. It was designed as an oral Topo II inhibitor and was evaluated in a breast cancer cell model in vitro and in vivo [185]. This compound showed great antiproliferative properties against the MCF7 and MDA-MB1 breast cancer cell lines, and it decreased the colony number of these cells in a dose-dependent manner. A kDNA-decatenation assay of compound **156** showed that it significantly inhibited the activity of DNA-Topo II in a concentration-dependent manner. At the same time, compound **156** also improved intracellular ROS levels and expression of the γ-H2AX protein, a DNA damage marker, whereas it arrested the MCF7 cell cycle in the S phase and increased the rate of apoptosis through activation of the mitochondrial pathway [185].

## 3. D-Ring Modifications of the Podophyllotoxin Skeleton

The *trans*-γ-lactone ring, or D-ring (Figure 3), of podophyllotoxin was initially considered essential for the cytotoxicity of cyclolignans and for the inhibition of tubulin polymerization [26]; however, it was found later that this ring was dispensable for the tubulin-binding activity of podophyllotoxin, although the Topo II inhibition ability can be influenced in some derivatives lacking this ring [16,186].

Our research group has been interested for many years in the design of podophyllotoxin derivatives lacking the lactone ring. In this regard, several podophyllic aldehyde derivatives hybridized with other bioactive compounds have been described, such as lignopurines [25] and more recently lignoquinones [187] and biscyclolignans [188].

Quinone and naphthoquinone derivatives are widely distributed in nature, and several compounds of this type have therapeutic properties. The in vitro cytotoxicity of naphthoquinones has been extensively studied [189,190]. Naphthoquinones undergo both one- and two-electron reduction, forming semiquinones or hydroquinones that can participate in different redox reactions. Terpenylquinones, on their part, are mixed biogenetic secondary metabolites widespread in nature that have interesting pharmacological activities, such as antifungal, antiviral, antileishmanial, or antineoplastic, among others [191]. Our research group also has a great deal of experience in the semisynthesis of this type of compound from natural terpenoids [192,193].

Lignohydroquinones are a family of molecular hybrids formed by the junction of the podophyllotoxin derivative, the podophyllic aldehyde, and several diterpenylnapthohydroquinones (DNHQs) and monoterpenylhydroquinones (MNHQs) through aliphatic or aromatic linkers [187,194] (Figure 45). The first class of compounds were synthesized by attaching the linker to the DNHQ (**160**) through the carboxylic acid presented in myrceocommunic acid (**157**) [188]. In the case of lignohydroquinones derived from MNHQs, oxidation of the β-myrcene rest double bond to carboxylic acid was first accomplished before linker attachment [194]. The cyclolignan fragment, on its part, was obtained from podophyllotoxin via opening of the *trans*-γ-lactone ring and subsequent protection of the two hydroxyl groups formed as an acetonide (**159**) before joining the hydroquinone rest through an ester bond. The last steps comprised deprotection of the acetonide and subsequent oxidation of the free hydroxyl groups under Swern conditions to obtain the α,β-unsaturated aldehyde, which is the main structural feature of podophyllic aldehyde. The antiproliferative activity of the new hybrids (**162**, **163**) was tested against three tumor cell lines. Among them, the DNHQ hybrid with the aromatic linker (**162**) showed the best antiproliferative activity against the MG-63 osteosarcoma cell line with an IC_50_ value of 0.15 µM. The results indicate a degree of selectivity of this compound toward the MG-63 cancer cell line. Compound **162** also had the ability to induce cellular apoptosis and cell cycle arrest in the G2/M phase in the MG-63 cell line, showing that the hybrid retained the tubulin effect associated with cyclolignans [187]. The NHQ-derived hybrid with the aromatic linker (**163**) was also found to be the most potent in its series. In this case, selectivity was found in the HT-29 line, where the hybrid reached the nanomolar IC_50_ range.

The 7β,9-biscyclolignan **166** (Figure 46) that was also synthesized by Castro’s group [188] was formed by two cyclolignan fragments joined through imine/amine-type bonds. One of the fragments was derived from 4′-*O*-demethylpodophyllotoxin, and the other was derived from podophyllic aldehyde. This hybrid was designed with the purpose to combine in a single structure the requirements to directly target both tubulin and Topo II. On one hand, the podophyllic aldehyde derivative (**164**) was synthesized via a procedure described previously by Castro et al. [26]. On the other hand, the fragment derived from 4′-*O*-demethylpodophyllotoxin (**165**) was synthesized in a one-pot reaction in which the 4′-*O*-demethylation and the 7-epimerization was carried out, and *p*-phenylenediamine acted as a linker to join both cyclolignans, as shown in Figure 46.

Cytotoxicity of this derivative (**166**) was evaluated at different concentrations against three human cell lines (MG-63, HT-29, and MCF7), and it exhibited remarkable activity against an osteosarcoma cell line (MG-63), even more than its precursors. Cell cycle and apoptosis studies showed that this biscyclolignan could induce apoptosis and cell cycle arrest at the G2/M phase in the MG-63 cell line. Finally, docking studies revealed that the new hybrid had the ability to interact with both tubulin and Topo II with good binding energies [188].

Other cyclolignans lacking the lactone ring described in the literature during the time covered by this review include hydrazide derivatives (**167a**–**j**, Figure 47) synthesized by Nerella and colleagues from podophyllotoxin and the corresponding anhydrous hydrazines [195]. Hydrazides have wide applications as chemical preservers for plants for manufacturing polymers and glues. In pharmacology, this chemical scaffold possesses a wide spectrum of bioactivity, including antibacterial, antitubercular, antifungal, anticancer, anti-inflammatory, anticonvulsant, antidepressant, antiviral, and antiprotozoal properties [196,197]. The antiproliferative activity of several podophyllotoxin hydrazide derivatives was evaluated. Compounds **167c** and **167f**, having ethyl and cyclohexyl substituents on the hydrazine, exhibited significant anticancer activity, almost similar to that of the standard drug etoposide, against all the cell lines tested. SAR studies revealed that compounds **167c** and **167f** showed the most promising apoptotic activity, while compounds with a benzyl moiety in hydrazine (compounds **167h**, **167i**, and **167j**) showed moderate activity [195].

Ganaie and colleagues synthesized C9′ triazolyl analogs of podophyllotoxin [198] while considering the importance of the triazole heterocycle in medicinal chemistry, as was mentioned previously in this article. The new derivatives were synthesized in a manner similar to that used by our own group some years before for several isooxazoline cyclolignans [199]. The triazole-containing podophyllotoxin derivatives (**171a**–**k**) were subsequently synthesized via Huisgen 1,3-dipolar cycloaddition with aromatic azides (Figure 48). Cytotoxicity of the new derivatives was tested against some human tumor cell lines. The results showed that compounds **171c**, **171e**, **171f**, and **171i** displayed better anticancer activity than the parent compound podophyllotoxin. The *para*-substituted derivatives **171e** and **171f** were the most promising derivatives of the series with IC_50_ values of 19 µM against the PC-3 prostatic tumor cell line for compound **171f** and 20µM against the A549 cell line for compound **171e** [198].

## 4. E-Ring Modifications of the Podophyllotoxin Skeleton

It has been observed that the maintenance of the free rotation of the E-ring (Figure 3) is necessary for antitumor activity. Additionally, a free 4′-hydroxy group is crucial in the DNA Topo II inhibitory activity of cyclolignans, and demethylation at this position seems to be necessary for DNA breakage activity to occur [14,16].

Due to the importance of the E-ring in the inhibition of DNA Topo II, E-ring modifications have only actually focused on enhancing E-ring involvement in the cytotoxic mechanism of the compound [12]. Frequently, modifications at this point complemented other changes, as occurred with some nitrophenylpiperazine derivatives prepared by Wu et al. [152] and mentioned previously in this article.

Wu and colleagues incorporated *O*-nitrophenoxyacetyl groups at the C4′ position with different substituents [152] (Figure 49). Derivative **172** was synthesized through the procedure described previously for the introduction of a piperazine scaffold at C7 (Figure 28) and then, final derivatives (**174**) were obtained (Figure 49). Most of the newly synthesized derivatives had antiproliferative activity against the tumor cell lines tested (the HeLa, A549, and HepG2 cell lines), but they showed the best potency in A549 and HeLa cells and the lowest potency in HepG2 cells. Compounds **174b** and **174d**, with 4-methyl and 4-methoxy groups as the R_1_, were the compounds that showed the best IC_50_ values of the series (0.77, 0.83, and 1.19 µM and 0.98, 0.91, and 1.58 µM against the A549, HeLa, and HepG2 cell lines, respectively). These compounds arrested the cell cycle in the G2/M phase in a dose-dependent manner and induced apoptosis in the HeLa cell line [152].

Compound **175** (Figure 18) was synthesized by Xiang and colleagues [200]. It combined deoxypodophyllotoxin and 5-FU, an antimetabolite widely used in the treatment of a wide range of cancers [201]. Compound **175** exhibited increased cytotoxicity in cancer cells and decreased toxicity in non-cancerous cells compared with etoposide. It also inhibited A549 cell migration via downregulation of MMP-9 expression and upregulation of TIMP-1 expression. It was synthesized using a procedure reported previously [202], but in the time covered by this review, further cytotoxic assays were made [200]. Interestingly, it was found recently that treatment with **175** resulted in a dramatic dose- and time-dependent inhibition of endothelial cell proliferation in low concentrations (20, 50, and 100 µM) compared to etoposide (30 µM) in normal human cells (WI-38). Also, compound **175** inhibited the wound-healing migration and the angiogenesis of umbilical vein endothelial cells (HUVECs) in vivo and in vitro in a dose-dependent manner [200].

Kwon et al., on their part, synthesized compound **178**, a new β-apopicropodophyllotoxin derivative [203]. β-Apopicropodophyllotoxin (**176**) was obtained from podophyllotoxin using *p*-toluensulfonyl chloride and cyclopentylamine. Then, it was 4′-*O*-demethylated using methanesulfonic acid (**177**), and finally, reaction with benzyl bromide yielded compound **178** (Figure 50). The cytotoxic activity of this compound was evaluated against the HTC 116 and DLD-1 colorectal cancer cell lines and the CCD-18CO human colon fibroblast cell line, showing less cytotoxicity against the CCD-18CO cell line and similar values of cytotoxicity against the colorectal cancer cell lines than podophyllotoxin and apopicropodophyllotoxin. This compound could induce DNA damage based on the increase in γH2AX levels, and it also showed radiosensitizer effects due to the increase in ROS levels in the HTC 116 and DLD-1 cells when treated with compound **178** in combination with radiation [203].

## 5. Conclusions and Future Perspectives

The present review shows that there is an ongoing search for new chemical entities containing podophyllotoxin in their structure.

Concerning the hybridization strategy, it should be noted that there is a wide variety of structures that have been conjugated with this natural product, including other natural products and different compounds of synthetic origin. Cytotoxic activity enhancement has been mainly pursued, but it is not the only strategy. In addition to scaffolds with antitumor activity, examples have been found of other molecules with activities that may synergize with cytotoxicity, such as redox ability or anti-inflammatory activities.

Regarding the chemical coupling of other scaffolds to podophyllotoxin, numerous examples were found. Most of the results presented in this review showed modifications of the C-ring at the C7 position of podophyllotoxin, mimicking the requirements and the mechanism of action of the drug etoposide. Etoposide is the reference molecule for the introduction of bulky substituents at such position; however, the binding strategies are not limited to that found in etoposide. The hydroxyl group at C7 has allowed linkage via oxygenated functional groups and also with other heteroatoms (N or S) at that position. Chemical strategies frequently used to introduce the second component of the hybrids include click chemistry, esterification, or substitution reactions, among others. The presence of heterocycles in the hybrid molecules is interesting and noteworthy, which motivated the organization of a single section within this review dedicated to systematizing the chemistry associated with these structures. Particularly, the use of 1,2,3-triazole systems obtained from 7β-azidopodophyllotoxin via click chemistry methodology is one of the most remarkable approaches. This heterocycle often serves as a bridge or linker for binding other types of molecules.

Other positions, such as D- and E-rings, for the formation of hybrids or conjugates have been less explored in this period. It might be interesting to explore these modifications further to chemomodulate podophyllotoxin by exploring its antimitotic profile, unlike the substitution at C7β, which leads to derivatives focused on Topo II inhibition.

According to the results presented in this review, it can be stated that podophyllotoxin is far from being just another compound of natural origin. It is nowadays a versatile and modulable lead compound in the search for new antitumor agents. Considering its properties and the possibilities of chemomodulation, it will be possible to continue searching for antitumor agents with better profiles and properties from this natural product.

Apart from the molecular hybridization described throughout this review, several other approaches recently have been gaining attention, such as bioconjugation or nanoconjugation. Nonetheless, such strategies can also be considered hybridization methods by which the natural product biopharmaceutical properties can be improved when increasing effectiveness of the natural product itself or promoting its targeted distribution and delivery [204]. These strategies well deserve to be the focus of a different review.

Moreover, podophyllotoxin is not only limited to its cytotoxic activity, although it seems to be the most emphasized activity in its chemomodulation. Its antiviral activity has long been well known [205,206]. In recent literature, examples of antiviral activity of podophyllotoxin derivatives different from the clinical indication against warts can be found [207]. In this regard, many authors have proposed podophyllotoxin [208] or etoposide [209] as useful drugs for the treatment of SARS-CoV-2 infection. In these cases, both compounds are used to ameliorate the symptoms associated with the inflammatory disease that accompanies the pathology. This is an example of a drug-repurposing approach. Furthermore, the immunomodulatory effect of these types of compounds [210,211] could be used to enhance the antitumor activity of other drugs or even be included as co-adjuvants in immunotherapy. Another perspective on the use of podophyllotoxin is in combination with other types of antitumor therapies. Thus, the radiodensity properties of podophyllotoxin and derivates have been described both in vitro and in vivo in studies of colon cancer and head and neck squamous carcinoma [203,212]. As described in this review, the combination in a single molecule with other compounds that can promote this activity could enhance said bioactivity and contribute to the repositioning of podophyllotoxin in combination with other therapeutic approaches.

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
