# Peer review of "Podophyllotoxin: Recent Advances in the Development of Hybridization Strategies to Enhance Its Antitumoral Profile"

_pharmaceutics, 2023, doi:10.3390/pharmaceutics15122728_

Round 1

Reviewer 1 Report

Comments and Suggestions for Authors

This review summarized the chemical modification of podophyllotoxin to improve the anti-tumor effect and decrease the side effect. Currently, there are no relevant articles summarizing and evaluating the cycliolignan skeleton of podophyllotoxin. Therefore, this article has significance. However, there are some suggestions to improve the quality of this article. 

1.     Authors mentioned in the abstract, “Not only are chemical modifications considered, but 20 also biological properties, including the use of diverse drug delivery systems”, however, there are no contents related to chemical modification with drug delivery system, please add.

2.     This paper is 63 pages long and contains a lot of information. Authors should add a table of contents.

3.     Contents should be more streamlined, and figures should be rearranged to make clearer and more concise.

4.     Full names and abbreviations of terms should comply with standards.

Comments on the Quality of English Language

English editing should be improved. 

Reviewer 2 Report

Comments and Suggestions for Authors

The manuscript discusses the recent advances in the development of hybridization strategies to enhance the antitumoral profile of podophyllotoxin. It focuses on the chemical modifications performed on the podophyllotoxin skeleton, including the synthesis of derivatives and their conjugation with other natural products or molecules. Based on the available content, the scientific quality of the manuscript appears to be satisfactory. I have the following brief comments:

1. The authors should consider providing more specific information in the abstract to give readers a clear understanding of the content and scope of the paper.

2. It would benefit from a more focused and concise presentation of the specific challenges associated with current treatments and the role of natural products, such as podophyllotoxin, in addressing these challenges.

3. how does different modifications contribute to different bioactivities? What do they target?

4. provide more details about molecular hybridization approach that the author mentioned in the conclusion as perspective.

Comments on the Quality of English Language

Need grammatical revision and proofreading

Reviewer 3 Report

Comments and Suggestions for Authors

This manuscript is a very interesting, thorough and exhaustive description and summarization of the state-of-the-art of podophyllotoxin-based drugs with improved anticancer properties.

This topic addresses a very crucial and pressing issue these days so it is worthy of presentation, the demonstration is original and relevant to the aimed research field.

This paper offer new and useful information compared to other review and research papers in the field and the methodology used is clear.

The Figures are also well designed informative and clear.

The Conclusion of the manuscript adequately summarizes and gives back the essence of the paper and the cited references are all appropriate.

Overall, this review paper provides a valuable contribution to the scientific literature

My minor comments on the paper as follows:

- Since the text of the paper is definitely large, placing a Table of contents is advisable at the beginning of the manuscript (after the abstract)

- At the end of the manuscript, adding a new section that covers the challenges of using podophyllotoxin chemicals including their practical applications as well as about the future perspectives on this field would increase the quality of paper even more.

Comments on the Quality of English Language

The quality of language is good.

Round 2

Reviewer 1 Report

Comments and Suggestions for Authors

Thanks for authors revision.